# Global patterns and drivers of soil microbial nitrogen and phosphorus use efficiency

Decai Gao[1], Yakov Kuzyakov[2,3], Manuel Delgado-Baquerizo [4], Josep Peñuelas [5,6], Daryl L. Moorhead[7], Robert L. Sinsabaugh[8], Xiaofeng Xu [9], Lifei Sun[10], Huimin Wang [1,11], Liang Kou [1,11], Xiaoli Fu[1,11], Xiaoqing Dai[1,11], Shengwang Meng[1,11], Ziping Liu[12], Siyu Wang[13], Frank Hagedorn [14], Matthias C. Rillig [15,16] & Yongxing Cui [15] ✉

While nutrient use efficiency of soil microorganisms, the proportion of assimilated nutrients allocated into biosynthesis rather than invested in mineralization, is a critical microbial functional trait, its global patterns remain poorly quantified. Here, we estimate microbial nitrogen use efficiency (NUE, $n = 2012$) and phosphorus use efficiency (PUE, $n = 3419$) across terrestrial ecosystems using the ecoenzymatic stoichiometric approach. Globally, NUE (mean 0.60) is nearly twice as high as PUE (0.35). Soil organic carbon (SOC) is the strongest predictor of both, with higher SOC associated with greater nutrient use efficiency. Spatial upscaling shows that tundra and boreal forest soils have markedly lower NUE than other regions, suggesting high nitrogen investments in nutrient acquisition in cold ecosystems, whereas PUE is similar across biomes, implying pervasively low phosphorus acquisition capacity. Our study identifies potential nutrient cycling hotspots worldwide and offers critical parameters to refine large-scale predictions of soil carbon and nutrient dynamics.

Nitrogen (N) and phosphorus (P) are the two most limiting nutrients for ecosystem productivity globally. To cope with these constraints, plants and soil microorganisms have evolved diverse mechanistic strategies to raise N and P use efficiency (i.e., NUE and PUE)[1,2]. Soil microorganisms break down high-molecular-weight organic compounds using ecoenzymes and assimilate the resulting low-molecular-weight organic N and P products[3,4]. Microbial NUE and PUE represent the proportion of these assimilated N and P allocated into biosynthesis (primarily growth) versus investment in mineralization pathways, reflecting the metabolic investment strategy between

[1]Qianyanzhou Ecological Research Station, Key Laboratory of Ecosystem Network Observation and Modeling, Institute of Geographic Sciences and Natural Resources Research, Chinese Academy of Sciences, Beijing, China. [2]Department of Soil Science of Temperate Ecosystems, Department of Agricultural Soil Science, University of Goettingen, Göttingen, Germany. [3]Peoples Friendship University of Russia (RUDN University), Moscow, Russia. [4]Laboratorio de Bio-diversidad y Funcionamiento Ecosistémico. Instituto de Recursos Naturales y Agrobiología de Sevilla (IRNAS), Consejo Superior de Investigaciones Científicas (CSIC), Av. Reina Mercedes 10, Sevilla, Spain. [5]CSIC, Global Ecology Unit CREAF-CSIC-UAB, Bellaterra , Catalonia, Spain. [6]CREAF, Cerdanyola del Vallès, Catalonia, Spain. [7]Department of Environmental Sciences, University of Toledo, Toledo, USA. [8]Biology Department, University of New Mexico, Albuquerque, USA. [9]Biology Department, San Diego State University, San Diego, CA, USA. [10]Key Laboratory of Environment Change and Resources Use in Beibu Gulf, Ministry of Education, and Guangxi Key Laboratory of Earth Surface Processes and Intelligent Simulation of Nanning Normal University, Nanning, China. [11]College of Resources and Environment, University of Chinese Academy of Sciences, Beijing, China. [12]Key Laboratory of Geographical Processes and Ecological Security in Changbai Mountains, Ministry of Education, School of Geographical Sciences, Northeast Normal University, Changchun, China. [13]School of Soil and Water Conservation, Beijing Forestry University, Beijing, China. [14]Soil Biogeochemistry, Swiss Federal Institute for Forest, Snow and Landscape Research WSL, Birmensdorf, Switzerland. [15]Institute of Biology, Freie Universität Berlin, Berlin, Germany. [16]Berlin-Brandenburg Institute of Advanced Biodiversity Research (BBIB), Berlin, Germany. ✉e-mail: cuiyongxing@zedat.fu-berlin.de

nutrient assimilation and potential mineralization[5–7]. Higher microbial NUE and PUE thus reflect more efficient allocation toward biomass production relative to mineralization investments[5,7]. Given their central importance in soil N and P cycling, microbial NUE and PUE are critical traits for understanding and predicting microbially mediated soil functions and nutrient dynamics. Yet, their global patterns and drivers remain largely unresolved[8,9], limiting the integration of microbial traits into microbial-based models of soil carbon (C) and nutrient cycling. This knowledge gap is particularly pressing given the growing anthropogenic disruption of global nutrient cycles, driven by climate and land-use change[10].

The global pattern and drivers of microbial NUE and PUE remain unclear for three main reasons. First, most existing measurements of microbial NUE and PUE are derived from incubation experiments with varied methods (i.e., $^{15}$N- $^{18}$O- and $^{33}$P-labeled substrate additions), environmental conditions (e.g., C availability, temperature, and moisture), and incubation durations (from hours to weeks)[5,11,12]. This methodological variability hinders the identification of consistent spatial patterns and environmental drivers. Second, isotope-based methods to measure microbial NUE and PUE are costly and logistically challenging to apply at large scales. As a result, the number of published studies using these methods remains limited, making it difficult to quantify global distributions of microbial NUE and PUE via data synthesis. Third, while stoichiometric models enable large-scale estimation of microbial NUE and PUE[13,14], their implementation requires multiple difficult-to-obtain parameters (e.g., microbial biomass, microbial activities, and soil nutrient supply), which has critically constrained their applications in large scales. Consequently, global assessments of microbial NUE and PUE based on existing approaches remain lacking.

Microbial nutrient use efficiency varies across terrestrial ecosystems, primarily due to widespread environmental and resource constraints in microbial metabolism[12,15]. Previous studies have revealed that NUE and PUE depend on biome type, climate, and soil properties, with higher efficiencies in forests than grasslands or croplands[5,15,16]. The availability and stoichiometric balance of nutrients in soils are likely major factors influencing microbial NUE and PUE[5,8,17], as microorganisms adjust metabolic strategies to maintain stoichiometric homeostasis. Competition between soil microorganisms and plants as well as between microorganisms for N and P further influence

microbial nutrient use strategies and efficiencies, with both groups dynamically reallocating limited resources in response to changing environmental conditions[4,18,19]. Consequently, anthropogenic nutrient inputs such as atmospheric N deposition may alter the magnitudes of microbial NUE and PUE[5,8,20]. Furthermore, organic C availability plays a critical role in microbial activity and growth[12,15,21], both soil organic carbon (SOC) and plant-derived C inputs could thus be key drivers of microbial NUE and PUE[16,22,23]. For example, incubation experiments suggest that sufficient C source supply (e.g., via labile C input) elevates microbial NUE and PUE by alleviating energy constraints[5,9,24], which explains high microbial NUE and PUE in forests, as forest soils typically have higher SOC content and C:N ratios than grasslands[25]. In contrast, climate conditions including temperature and precipitation are also important regulators of microbial activity and vary across biomes, which may lead to biome-specific difference in microbial NUE and PUE. However, it is unclear whether high microbial NUE and PUE in forests are primarily driven by C availability or by climate, and which factors act as the key drivers of their global patterns.

Here, we hypothesized high microbial NUE in boreal/tundra ecosystems due to widespread N limitation in cold regions, whereas NUE could be low in the tropics, where rapid mineralization and leaching reduce microbial N retention. Microbial PUE could follow an inverse pattern with the high value in P-limited tropical soils but a low value in cold regions where P limitation is less pronounced. To address this knowledge gap and test our hypotheses, we compile a global dataset comprising 2,012 and 3,419 paired observations for estimating microbial NUE and PUE, respectively, using the ecoenzymatic stoichiometry model. Specifically, these observations include variables related to soil nutrient availability, microbial biomass, and extracellular enzymes activities in soils, extracted from 213 published studies (Fig. 1). To ensure consistency and comparability across studies, we only include datasets where microbial biomass is determined using the chloroform fumigation-extraction method, and extracellular enzyme activities were measured based on fluorometric substrate assays, which are widely adopted standard protocols in soil microbial ecology. The estimated microbial NUE and PUE reflect the metabolic investment strategy, as reflected in microbial allocation toward C- *vs.* N- or P-acquiring ecoenzymes, linking microbial nutrient demand with soil nutrient availability. They thus capture microbial trade-offs in resource partitioning toward growth versus potential nutrient

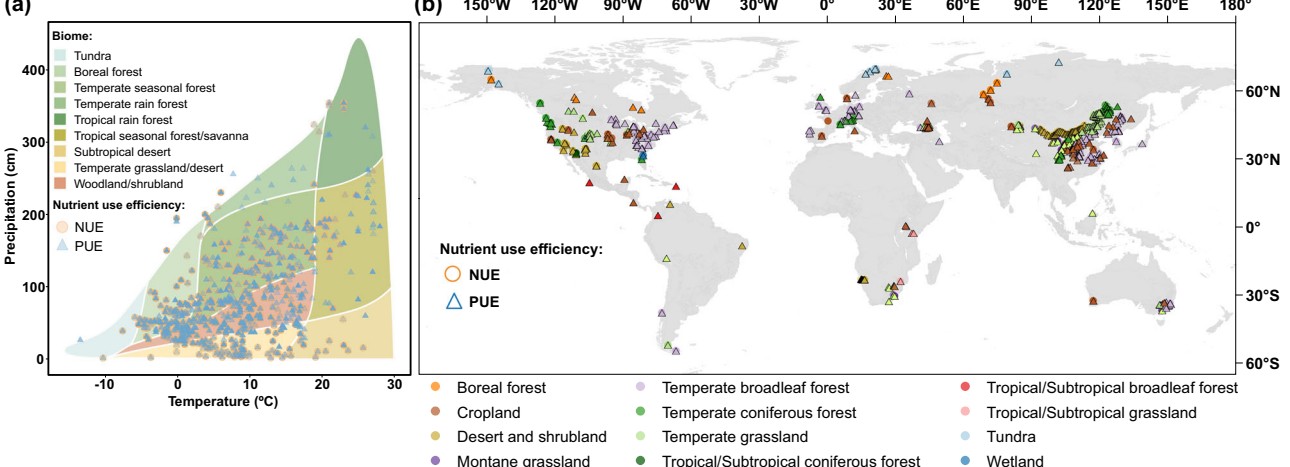

**Fig. 1 | Global distribution of sampling sites of observations for estimating microbial nutrient use efficiency in soil. a** Distribution of sampling sites in climate space following the Whittaker biome framework, defined by mean annual temperature (MAT, °C) and mean annual precipitation (MAP, cm). Each point represents an individual observation included in the dataset. **b** Geographic distribution of sampling sites included in this study. Circles represent microbial nitrogen use efficiency (NUE), and triangles represent microbial phosphorus use efficiency (PUE). Colors indicate biome categories assigned according to the MODIS land cover classification (IGBP scheme). Coordinates are displayed in geographic projection (WGS84). Gray areas indicate regions without sampling data.

**Table 1 | Predicted microbial nitrogen use efficiency (NUE) and phosphorus use efficiency (PUE) across soils of major biomes based on Random Forest models**

| Biome | Microbial NUE | Microbial PUE |
|---|---|---|
| Boreal forest | 0.53 ± 0.08e | 0.37 ± 0.07a |
| Montane grassland | 0.61 ± 0.05c | 0.33 ± 0.04 d |
| Temperate broadleaved forest | 0.62 ± 0.07c | 0.36 ± 0.05ab |
| Temperate coniferous forest | 0.62 ± 0.05c | 0.35 ± 0.04bc |
| Temperate grassland | 0.59 ± 0.06 d | 0.33 ± 0.04 d |
| Tropical/subtropical broadleaved forest | 0.68 ± 0.04a | 0.37 ± 0.05a |
| Tropical/subtropical coniferous forest | 0.69 ± 0.04a | 0.34 ± 0.03 cd |
| Tropical/subtropical grassland | 0.66 ± 0.04b | 0.34 ± 0.03 cd |
| Tundra | 0.48 ± 0.08 f | 0.33 ± 0.03 d |
| Cropland | 0.61 ± 0.08c | 0.34 ± 0.04 cd |
| Wetland | 0.59 ± 0.10 d | 0.36 ± 0.05ab |
| Global mean | 0.60 | 0.35 |

Values are presented as mean ± standard deviation (SD). Different lowercase letters within each column denote statistically significant differences among biomes (one-way ANOVA with Tukey's post-hoc test, $p < 0.05$).

mineralization, rather than measuring actual in-situ process rates. We further applied Random Forest models incorporating 12 environmental variables related to climate, vegetation, and soil factors to identify key drivers and upscale predictions of microbial NUE and PUE at 1 km × 1 km spatial resolution. Our study presents the global assessment of microbial NUE and PUE, offering critical insights into soil N and P cycling and their potential responses to environmental change.

## Results and Discussion
### Key drivers of microbial NUE and PUE worldwide

The estimated global mean values were 0.60 for NUE and 0.35 for PUE, respectively (Table 1). Among four linear and four nonlinear models tested, Random Forest models yielded the strongest prediction accuracy for microbial NUE ($R^2 = 0.75$) and PUE ($R^2 = 0.82$) (Fig. 2 and Supplementary Table 1). SOC content emerged as the most important factor influencing microbial NUE and PUE at the global scale (Fig. 2). Partial regression analysis indicated that both efficiencies increased markedly with rising SOC levels in soils with low initial SOC content ($p < 0.05$; Figs. 3 and 4), likely because added C alleviates energy limitation and stimulates microbial investment in nutrient acquisition via enzyme productions. However, this positive trend plateaued under relatively low to medium SOC levels (Figs. 3 and 4), suggesting a shift from energy to nutrient limitation. In these environments, microbial nutrient use efficiency may be constrained by the availability of N or P rather than C, highlighting interacting stoichiometric constraints on microbial metabolism. In low-SOC soils where labile C pools are typically depleted, exogenous labile substrate addition supplies readily available C and energy to support microbial metabolism, enabling more efficient assimilation of N and P into biomass[24]. This reduces nutrient losses via excretion and mineralization, thereby increasing microbial NUE and PUE[5]. SOC could also favor the production of extracellular enzymes such as β-glucosidase (BG), N-acetyl-glucosaminidase (NAG), and acid phosphatase (AP), all of which are essential for nutrient acquisition[14]. Under C-rich conditions, soil microorganisms prioritize nutrient retention to maintain cellular stoichiometric homeostasis[5,20,26]. The surplus energy from SOC allows greater investment in assimilatory processes relative to dissimilatory

pathways, thereby raising NUE and PUE by increasing biomass growth. These SOC-driven mechanisms are further supported by positive correlations between microbial NUE and PUE and the normalized difference vegetation index (NDVI) ($p < 0.05$; Figs. 3 and 4), as higher NDVI reflects increased plant-derived C inputs[22,27]. Further, with increases in C availability, limited nutrient availability may become a restrict in NUE and PUE, reflecting a transition from energy to nutrient limitation. These findings underscore the dosage-effect of SOC in regulating microbial nutrient use efficiency (Fig. 3c).

Climatic factors, including humidity, mean annual temperature (MAT), and potential evapotranspiration (PET), also played critical roles in shaping global patterns of microbial NUE and PUE (Fig. 2). Temperature and precipitation jointly influence the balance between microbial biomass synthesis and the mineralization of N and P[15]. Below thermal optima (~25 °C for fungi, ~35 °C for bacteria), increasing temperature generally increases microbial growth rates[28,29]. This raises greater allocation of N and P toward anabolic processes, thereby elevating microbial NUE and PUE. Soil moisture regulates microbial metabolism bidirectionally: under mild to moderate dry conditions, rising moisture typically increases microbial NUE and PUE by alleviating water limitation for microorganisms and nutrient diffusion constraints[15,30]. In contrast, high moisture can cause oxygen depletion and hypoxic stress, which may reduce microbial growth and nutrient use efficiency[15,31]. These results illustrate the sensitivity of microbial nutrient use efficiency to climate change and highlight climate-mediated shifts in microbial allocation strategies across ecosystems.

As an indicator of the water-energy balance, high PET is generally associated with increased aridity and temperature, often leading to reduced microbial metabolic rates[32]. However, microbial NUE and PUE displayed fluctuant responses to PET (Figs. 3 and 4). Specifically, NUE showed an initial transient increase followed by a rapid decline, secondary rise, and subsequent stabilization at high PET levels. In contrast, PUE increased initially before declining and stabilizing at elevated PET. The declining phases under higher PET reflect rising microbial stress due to increased aridity and temperatures that typically induce C limitation and potentially mineral N accumulation. Under these PET-related conditions, microbial communities may shift metabolic priorities from growth to maintenance, reducing nutrient use efficiency. Specifically, microbial NUE decreases as C scarcity forces microorganisms to prioritize catabolic adenosine triphosphate (ATP) generation for cellular maintenance, while substantial N resources are diverted toward the production of C-acquiring ecoenzymes[13,15,33]. The stabilization of microbial NUE and PUE at relatively high PET levels may reflect microbial acclimation or a shift in community composition. Microbial communities may adopt conservative strategies, such as reducing extracellular enzyme production and increasing intracellular N recycling, to cope with persistent resource imbalances[5]. These adaptations involve changes in enzymatic profiles or adjustments in biomass stoichiometry, allowing microorganisms to sustain metabolic efficiency under environmentally stressful conditions.

### Global hotspots of microbial NUE and PUE

We mapped the global distribution of NUE and PUE by integrating their estimates with 11 global-scale environmental covariates related to climate, soil properties, and vegetation. Using Random Forest models, we generated spatially explicit predictions of microbial NUE and PUE (Fig. 5). Modeled microbial NUE values showed a clear latitudinal gradient, decreasing by 22% from tropical to boreal regions (Fig. 5 and Table 1). Specifically, the highest mean values occurred in tropical/subtropical regions (0.66–0.69), followed by temperate regions (0.59–0.62), with the lowest values in boreal regions (0.53). Among biomes, tropical/subtropical coniferous forests had the highest microbial NUE (0.69), closely followed by tropical/subtropical

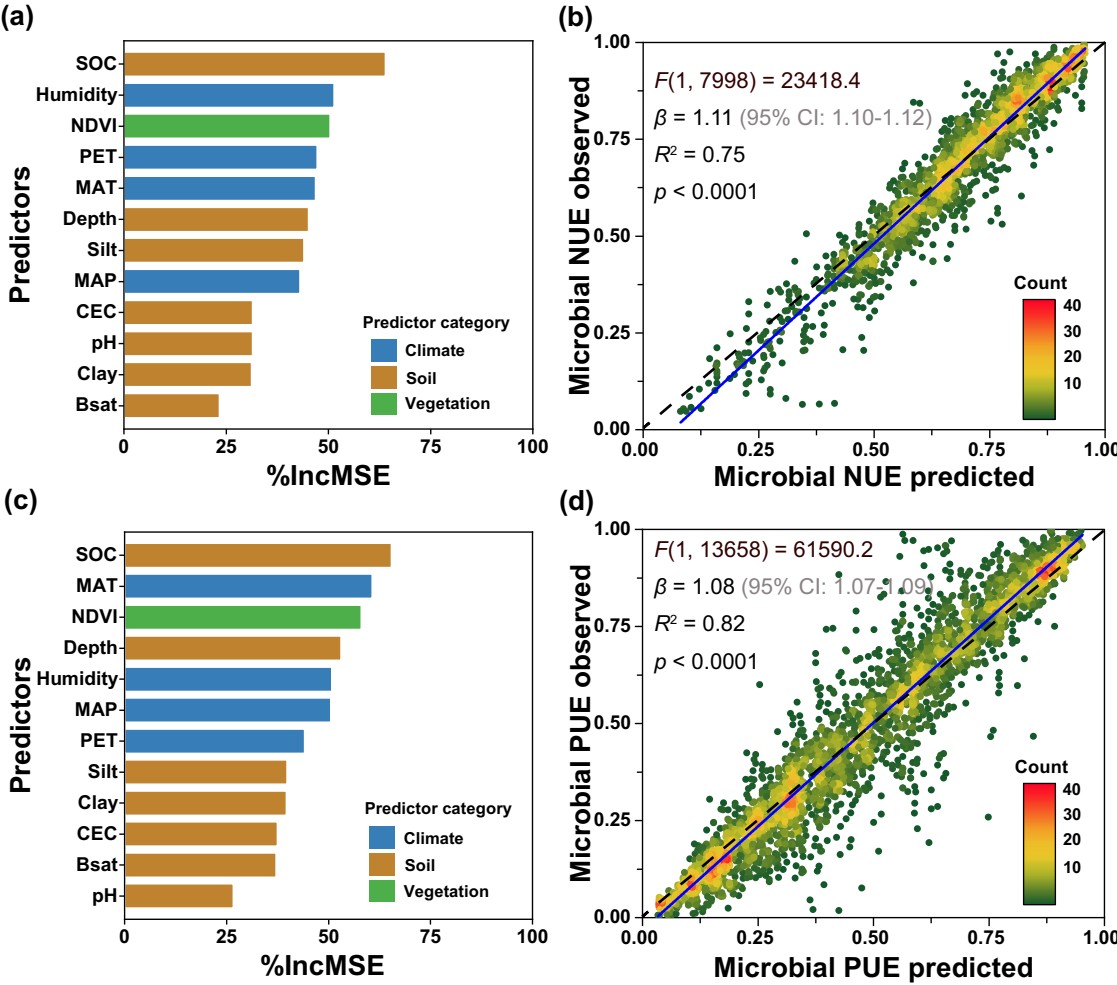

**Fig. 2 | Relative importance of climatic, vegetation, and edaphic factors for microbial nutrient use efficiency and model performance. a** Relative importance of climatic, vegetation, and soil factors for microbial nitrogen use efficiency (NUE). **b** Predicted *vs.* observed microbial NUE. **c** Relative importance of climatic, vegetation, and soil factors for microbial phosphorus use efficiency (PUE). **d** Predicted *vs.* observed microbial PUE. In **b** and **d**, the solid blue line represents the fitted linear regression between predicted and observed values, and the black dotted line denotes the 1:1 relationship. The green-to-red gradient indicates data amount (0–40). In **a** and **c**, blue, brown, and green represent climatic, soil, and vegetation predictors, respectively. Climatic variables include mean annual precipitation (MAP, mm), mean annual temperature (MAT, °C), potential evapotranspiration (PET, mm), and average annual relative humidity (Humidity, %). Soil variables include soil organic carbon (SOC, %), soil pH (pH), soil silt content (Silt, %), soil clay content (Clay, %), cation exchange capacity (CEC, cmol kg$^{-1}$), base saturation (Bsat, %), and soil depth (Depth, cm). The vegetation variable is normalized difference vegetation index (NDVI). Variable importance in **a** and **b** was quantified using percent increase in mean squared error (%IncMSE), representing the increase in prediction error when a predictor was permuted. Exact one-sided permutation *p*-values (*n* = 500 permutations) are provided in Supplementary Table 3. For NUE, the Random Forest models yielded a mean squared residual of 0.0093 and explained 76.35% of the variance. For PUE, the mean squared residual was 0.0135, with 82.65% of variance explained.

broadleaved forests (0.68). In contrast, boreal forests (0.53) and tundra ecosystems (0.48) displayed the lowest values.

Contrary to our hypothesis that microbial NUE could be higher at high-latitude regions than others, our results revealed lower values in these regions (Fig. 5 and Table 1). These predictions, based on the ecoenzymatic stoichiometric approach, suggest that microorganisms allocate a greater portion of assimilated N to enzyme production for organic matter decomposition. This aligns with the concept that microbial NUE is governed not simply by N availability or temperature constraints, but by a combination of competitive dynamics and microbial resource allocation strategies. In nutrient-limited environments, microbial communities (e.g., ectomycorrhizal fungi) prioritize enzymatic investment for decomposing recalcitrant organic matter over biomass production[26,34]. This C and energy reallocation diverts assimilated N from growth to enzyme synthesis, explaining the observed reduction in NUE. The resulting mobilization of organically bound N may partially alleviate N limitation despite ecosystem-level

scarcity. Moreover, the fungal dominance in high-latitude biomes, characterized by a high C:N ratio and lower stoichiometric differences from recalcitrant substrates such as lignin-rich litter, may further reduce microbial N demand and contribute to lower NUE[26]. Accordingly, the lower NUE in high-latitude biomes may reflect microbial strategies to adapt to N-poor conditions and is therefore consistent with the general view that these cold ecosystems are primarily N-limited.

The observed decline in microbial NUE with increasing latitude (Fig. 5a) contrasts with the well-documented latitudinal increase in microbial CUE[19,35,36], revealing a microbial trade-off between these traits across climatic gradients. In tropical ecosystems, soils experience substantial P limitation while maintaining tight N cycling[19,37]. Despite rapid N mineralization rates, microorganisms prioritize N retention through efficient immobilization-recycling pathways to counter intense plant competition and leaching losses[19,38–40], resulting in higher NUE (Fig. 5a). Moreover, elevated respiratory costs at high

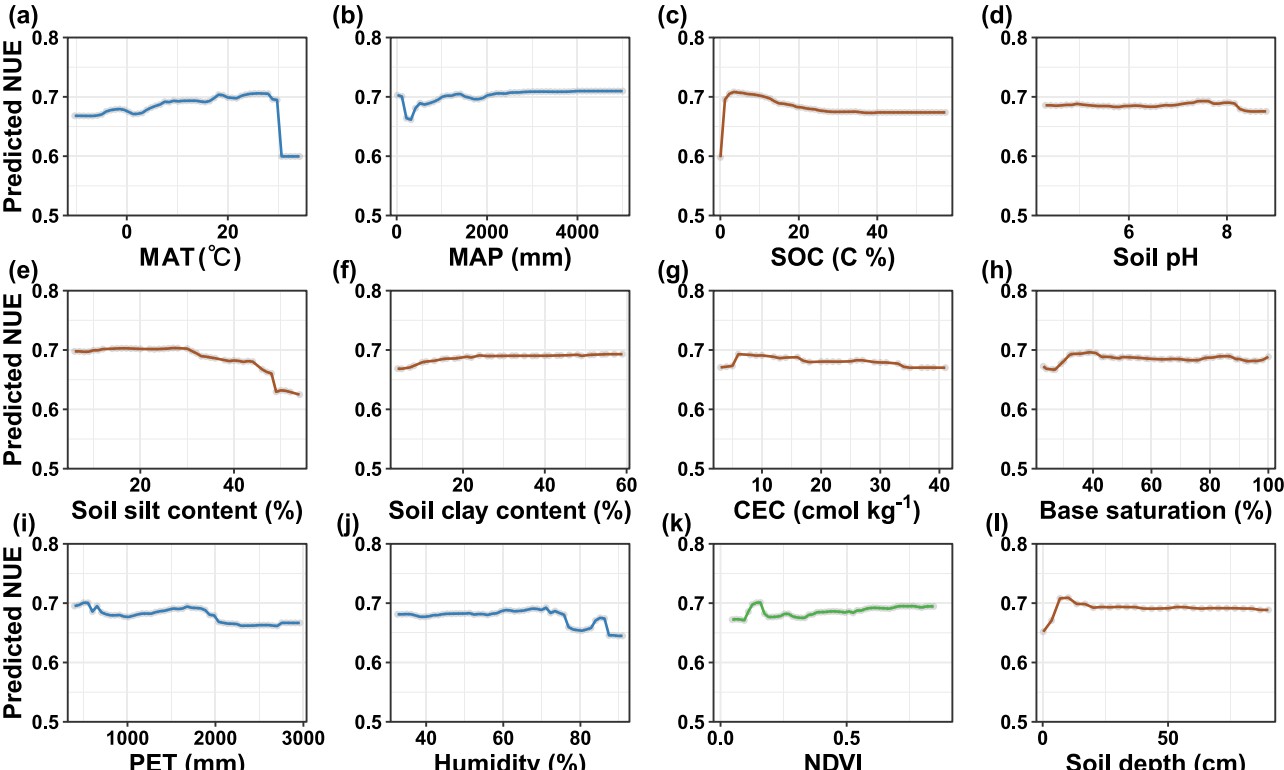

**Fig. 3 | Partial dependence plots showing the relationships between microbial nitrogen use efficiency (NUE) and environmental predictors.** Panels show the marginal effects of individual predictors on NUE: **a** mean annual temperature (MAT, °C); **b** mean annual precipitation (MAP, mm); **c** soil organic carbon (SOC, %); **d** soil pH; **e** soil silt content (%); **f** soil clay content (%); **g** cation exchange capacity (CEC, cmol kg⁻¹); **h** base saturation (Bsat, %); **i** potential evapotranspiration (PET, mm); **j** mean annual relative humidity (Humidity, %); **k** normalized difference vegetation index (NDVI); and **l** Soil depth (cm). Blue, brown, and green lines represent climatic, soil, and vegetation predictors, respectively.

temperatures reduce C retention, thereby lowering microbial CUE. In boreal and tundra ecosystems, the scenario is reversed. Cold temperatures slow organic matter decomposition, favoring C immobilization and thus higher microbial CUE[35]. However, microbial NUE derived from enzymatic stoichiometry declines because energy allocation is prioritized toward the production of extracellular enzymes for decomposition of recalcitrant organic matter rather than microbial biomass synthesis[26,34], diverting assimilated N away from growth. Additionally, pulsed N release during thaw events can create transient rises in N availability exceeding microbial demand[41]. Thus, while CUE is largely governed by temperature constraints on metabolic efficiency, NUE could be influenced more strongly by microbial nutrient acquisition strategies and competitive dynamics.

In contrast to microbial NUE, microbial PUE did not show a clear latitudinal trend ($p > 0.05$; Fig. 5 and Table 1). This lack of apparent global pattern is likely attributed to three main reasons: (i) the pervasive P limitation across biomes, (ii) the inherently slower and more variable cycling of P compared to N, which represses microbial capacity to regulate P use, and (iii) dominant local-scale controls such as soil age, mineralogy, and land use[19,42]. Moreover, specific latitudinal drivers may offset each other; for instance, geochemical P retention in tropical soils may counterbalance temperature-driven stimulation of mineralization, while lower geochemical P limitation in high latitudes is partially offset by slower organic matter turnover. However, we found some regional hotspots of microbial PUE, such as a high-PUE region in North America (Fig. 5b), which could result from the combination of strong P limitation of coniferous litter and strong P sorption via metal oxides, which promotes highly efficient microbial P conservation strategies[43]. These findings suggest that microbial PUE is primarily governed by local-scale factors, including soil type, vegetation composition, and microsite climatic conditions. This resulting high variability in P cycling from such localized heterogeneity may obscure continental or global-scale environmental gradients[44], which reflects fundamental differences in how microbial stoichiometric homeostasis operates for P vs. N.

Forests had higher microbial NUE and PUE than grasslands, averaging 3.2% greater in tropical/subtropical zones and 8.9% greater in temperate zones ($p < 0.05$; Table 1). This could be attributed to forest soils typically have higher C:N and C:P ratios due to lignin-rich plant litter (both foliar and root-derived) compared to grassland residues[45,46]. These stoichiometric imbalances increase microbial demand for N and P to maintain stoichiometric homeostasis. When substrate C:nutrient ratios exceed microbial metabolic thresholds, microorganisms initially prioritize ecoenzyme production over biomass retention, increasing nutrient acquisition at the cost of growth, thereby decreasing NUE and PUE[47]. While high substrate C:nutrient ratios force investment in ecoenzymes (a metabolic cost), this strategy allows microbes to minimize nutrient losses, thereby maintaining high cellular N and P retention (high NUE and PUE) despite the energetic cost to C-acquisition (low CUE). Although forests may harbor more diverse and functionally redundant microbial communities[48], enzymatic strategies appear primarily governed by stoichiometric demands rather than diversity per se. For instance, fungal-dominated communities in forests efficiently decompose high C:N ratio litter by directing limited N toward ligninolytic enzyme synthesis, thereby optimizing nutrient acquisition under chronic N limitation[26]. This adaptation is crucial for resilience to global change: Under projected increases in atmospheric $CO_2$ (which can further raise plant tissue C:N ratios) and warming (which can alter decomposition kinetics), the high nutrient-use efficiency of forest ecosystems may buffer against

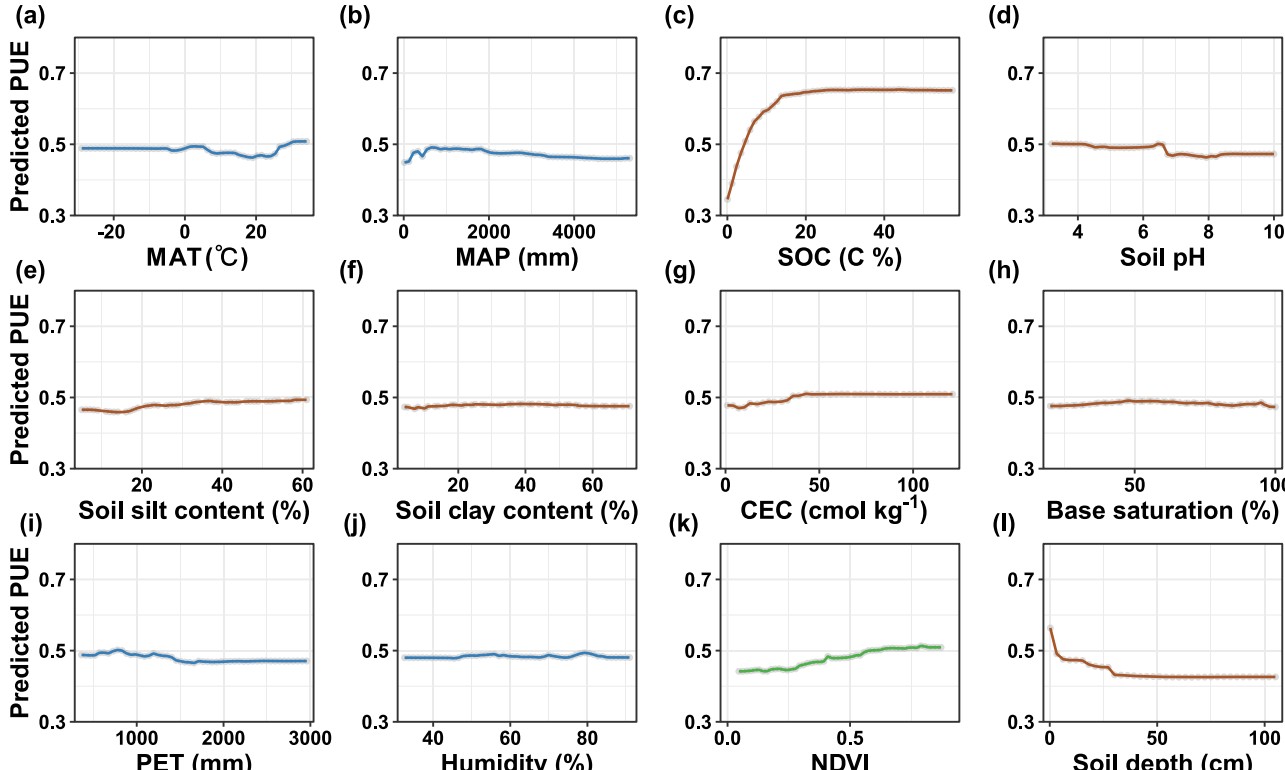

**Fig. 4 | Partial dependence plots showing the relationships between microbial phosphorus use efficiency (PUE) and environmental predictors.** Panels show the marginal effects of individual predictors on PUE: **a** mean annual temperature (MAT, °C); **b** mean annual precipitation (MAP, mm); **c** soil organic carbon (SOC, %); **d** soil pH; **e** soil silt content (%); **f** soil clay content (%); **g** cation exchange capacity (CEC, cmol kg⁻¹); **h** base saturation (Bsat, %); **i** potential evapotranspiration (PET, mm); **j** mean annual relative humidity (Humidity, %); **k** normalized difference vegetation index (NDVI); and **l** Soil depth (cm). Blue, brown, and green lines represent climatic, soil, and vegetation predictors, respectively.

nutrient losses. Grasslands, with their lower baseline NUE and PUE driven by lower C:nutrient ratios, may be more susceptible to nutrient leaching losses under warming-induced mineralization pulses, potentially reducing long-term C sequestration potential. Consequently, quantifying the specific microbial N and P use strategies in forests and grasslands will provide a vital mechanistic basis for modeling how these major biomes regulate nutrient retention, C storage, and greenhouse gas feedback under future climate scenarios.

## Uncertainties and implications

These findings provide important insights into the global patterns and drivers of microbial NUE and PUE, but they are limited by several aspects. First, the ecoenzymatic stoichiometric approach used here relies on extracellular enzyme activities (BG, NAG, LAP, and AP) as proxies of microbial nutrient acquisition, assuming that enzyme activity ratios reflect imbalances between microbial nutrient demands and soil nutrient availability[14,49]. Correspondingly, the approach simplifies the complexity of microbial processes by focusing on a limited enzyme set and does not fully capture variation in enzyme kinetics, microbial community composition, or biotic interactions such as mycorrhizal symbioses. It is also important to note that differences in methodological approaches, such as ecoenzymatic stoichiometry versus isotopic tracers, can lead to divergent interpretations of microbial nutrient use efficiency[50]. For instance, Sun et al.[50] found opposing latitudinal trends for microbial NUE, by comparing the two prevalent methods (i.e., ecoenzymatic stoichiometric approach, as employed in our study, and the ¹⁸O-based isotopic approach). In fact, these methods capture different microbial processes, extracellular investment strategies versus intracellular assimilation efficiency, which can reflect contrasting ecological patterns. Compared to the ¹⁸O

method, the ecoenzymatic stoichiometric approach focuses on the enzymatic investment strategy employed by microorganisms in response to stoichiometric imbalances between microbial biomass and external substrates, emphasizing the enzymatic investment strategy to acquire limiting resources rather than directly quantifying mineralization fluxes[14]. Therefore, future efforts to develop integrated approaches are essential to reconcile these perspectives and advance mechanistic understanding of microbial resource use.

Second, despite the global scope of our dataset, spatial heterogeneity in sampling intensity introduces uncertainty, particularly in tropical and boreal ecosystems where observation sites are sparsely distributed (Fig. 1). Therefore, prioritizing studies in these underrepresented regions would strengthen the robustness of future research. Third, we did not consider the effects of plant-microbe interactions on microbial NUE and PUE. For example, nutrient competition between plants and microorganisms could reduce microbial NUE and PUE, whereas collaborations (e.g., symbiotic relationships between plants and fungi) could increase microbial NUE and PUE. Lastly, a limitation arises from using SOC as a predictor for global NUE and PUE estimates, because SOC was also used to estimate NUE and PUE in the stoichiometry models. Despite VIF screening (threshold ≤ 5) to reduce multicollinearity, residual correlations may persist, slightly overestimating the predictive importance of SOC. However, the SOC used for estimating NUE and PUE is derived from our data synthesis of published studies, whereas the SOC used as an environmental predictor is obtained from a public database (Supplementary Table 2). This distinction helps minimize potential collinearity issues. Despite these caveats, our cross-validation analyses show that ecoenzymatic stoichiometric modeling produces robust and scalable estimates of microbial NUE and PUE at the global scale.

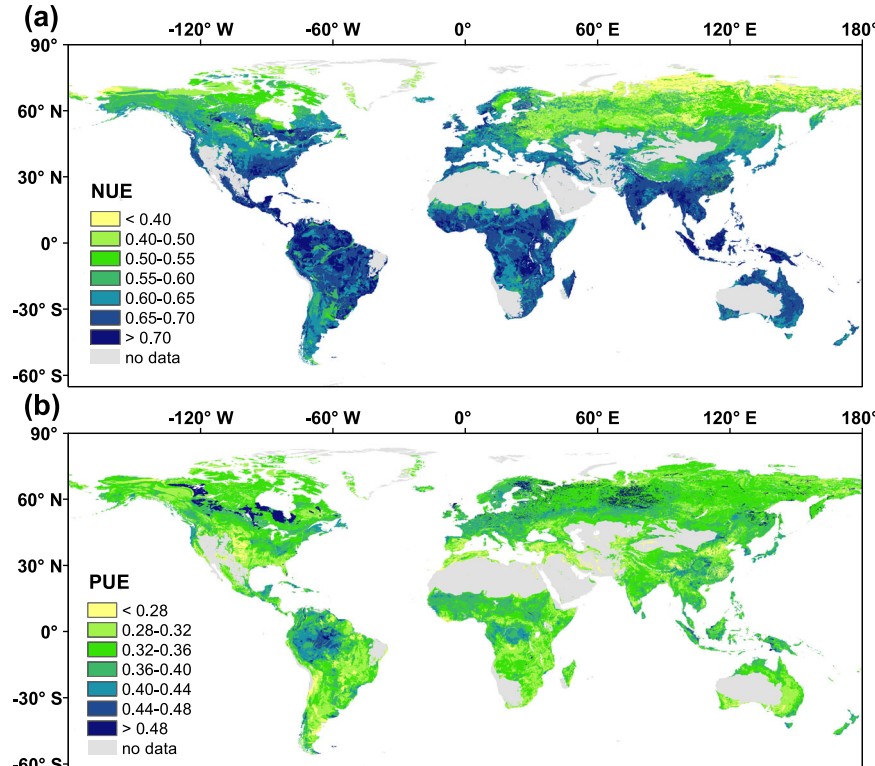

**Fig. 5 | Global upscaled patterns of microbial nutrient efficiency in soil predicted by Random Forest models. a** Global distribution of predicted microbial nitrogen use efficiency (NUE) in soil. **b** Global distribution of predicted microbial phosphorus use efficiency (PUE) in soil. Spatial predictions were generated using Random Forest models at a spatial resolution of 1 km × 1 km. Relative prediction uncertainties for NUE and PUE are presented in Supplementary Figs. 1, 2.

In summary, our study reveals global patterns and key drivers of soil microbial NUE and PUE, offering insights into microbially mediated soil nutrient cycling. Microbial NUE showed a clear latitudinal gradient, peaking in tropical/subtropical regions and declining in boreal regions, driven primarily by contrasting microbial energy allocation strategies and N retention mechanisms influenced by macroclimate patterns. In contrast, Microbial PUE showed no consistent latitudinal trend, suggesting stronger roles of localized factors such as soil nutrient availability in microbial P use, and microbial stoichiometric flexibility in adjusting to highly variable soil C:P ratios. SOC emerged as the strongest predictor of both microbial NUE and PUE, emphasizing the central role of C availability in regulating microbial nutrient use strategies. Forest ecosystems consistently had higher microbial NUE and PUE than grasslands, likely due to higher recalcitrant organic matter in forest soils that increases microbial demand for nutrients and drives enzyme-mediated nutrient foraging. Together, these findings highlight the divergent effects of soil properties, climate, and vegetation in shaping microbial nutrient use efficiency across ecosystems. Future research should prioritize disentangling local- and global-scale drivers of microbial NUE and PUE and evaluating how global change factors, such as warming and altered precipitation regimes, affect microbial nutrient dynamics. These insights are essential for improving ecosystem models and informing sustainable land management practices to support long-term soil fertility and ecosystem functioning.

## Methods
### Data source and processing
Data were extracted from previously peer-reviewed articles published up to 2022, sourced from the Web of Science and Google Scholar databases. The search was conducted using the following keywords: "extracellular enzyme" OR "enzymatic activity" OR "soil enzyme activity". The criteria for data selection were as follows: (i) studies were included if they reported data on β-glucosidase (BG), N-acetyl-glucosaminidase (NAG), leucine-amino-peptidase (LAP), and acid phosphatase (AP) together; (ii) data on soil extracellular enzymatic activity were determined by the fluorogenic substrate method. A total of 213 papers met the selection criteria, yielding 2,012 and 3,419 observations of microbial NUE and PUE, respectively (Supplementary Data 1).

For each study site, we compiled detailed information, including geographic coordinates (latitude and longitude), biome type, climate variables (mean annual temperature (MAT) and mean annual precipitation (MAP)), soil properties (SOC, total nitrogen (TN), total phosphorus (TP)), and soil microbial biomass (microbial biomass carbon (MBC), microbial biomass nitrogen (MBN), microbial biomass phosphorus (MBP)). When original studies did not fully report all soil nutrient and microbial biomass parameters, missing indicators were extracted from global data maps using the geographical coordinates of the sampling sites. SOC and TN contents were extracted from SoilGrids database (https://soilgrids.org)[51]. A global map of TP content was taken from the Land-Atmosphere Interaction Research Group at Sun Yat-sen University (http://global-change.bnu.edu.cm/research/soilw)[52]. Global maps of MBC, MBN, and MBP contents were from a previous study[53]. Biome classifications were derived from three primary sources: global cropland data, available at http://www.earthstat.org/cropland-pasture-area-2000/; global wetland data, accessible at https://data.giss.nasa.gov/landuse/wetland.html; and data for other biomes derived from a vegetation map, found at https://services.arcgis.com/BG6nSlhZSAWtExvp/arcgis/rest/services/GlobalBiomes/FeatureServer. For biome-level analysis, data were categorized into eleven distinct terrestrial biomes: boreal forest, temperate broadleaved forest, temperate coniferous forest, tropical/subtropical broadleaved forest, tropical/subtropical coniferous

forest, montane grassland, temperate grassland, tropical/subtropical grassland, tundra, wetland, and cropland. The global distribution of sampling sites is depicted in Fig. 1. Notably, Antarctica was not included in this analysis, so the study does not represent a truly global coverage.

## Data analysis

Soil microbial NUE and PUE were calculated using an ecoenzymatic stoichiometric approach based on soil enzyme activities[13,14]. This approach provides a framework for understanding how microorganisms allocate resources to acquire C, N, and P in response to environmental nutrient availability. Microbial extracellular enzymes are critical for the breakdown of organic matter and the release of nutrients for microbial uptake. The activities of these enzymes reflect microbial nutrient demands and limitations. The ecoenzymatic stoichiometric approach uses ratios of enzyme activities involved in C, N, and P acquisition to infer microbial nutrient-use efficiency. Specifically, microbial NUE and PUE are calculated based on the balance between microbial biomass composition and available soil nutrients, as mediated by enzyme activities. The equations used for estimation were as follows:

$$NUE_{N:C:P} = NUE_{max}\{(S_{N:C}S_{N:P})/[(S_{N:C} + K_{N:C}) \times (S_{N:P} + K_{N:P})]\}^{0.5} \quad (1)$$

$$PUE_{P:C:N} = PUE_{max}\{(S_{P:C} \times S_{P:N})/[(S_{P:C} + K_{P:C}) \times (S_{P:N} + K_{P:N})]\}^{0.5} \quad (2)$$

where $NUE_{max}$ and $PUE_{max}$ represent the theoretical maximum values for NUE and PUE, respectively. Both $NUE_{max}$ and $PUE_{max}$ are set to 1, reflecting the assumption that microbial nutrient use efficiency cannot exceed 100%. This upper limit is based on the theoretical framework of nutrient assimilation efficiency[13,14]. The half-saturation constants $K_{N:C}$, $K_{N:P}$, $K_{P:C}$, and $K_{P:N}$ are set to 0.5, representing the enzyme activity ratios at which microbial nutrient acquisition reaches half of its maximum potential. These values are empirically derived and reflect the balance between enzyme activity and nutrient availability[13,14]. The scalars $S_{N:C}$, $S_{N:P}$, $S_{P:C}$, and $S_{P:N}$ are used to quantify the balance between microbial biomass stoichiometry and available nutrients, normalized by enzyme activity ratios. These scales are calculated as follows:

$$S_{N:C} = (1/EEA_{N:C})(B_{N:C}/L_{N:C}) \quad (3)$$

$$S_{N:P} = (1/EEA_{N:P})(B_{N:P}/L_{N:P}) \quad (4)$$

$$S_{P:C} = (1/EEA_{P:C})(B_{P:C}/L_{P:C}) \quad (5)$$

$$S_{P:N} = (1/EEA_{P:N})(B_{P:N}/L_{P:N}) \quad (6)$$

Where $EEA_{N:C}$ is the ratio of the sum of NAG and LAP to BG ((NAG + LAP)/G); $EEA_{N:P}$ is the ratio of the sum of NAG and LAP to AP ((NAG + LAP)/AP); $EEA_{P:C}$ is the ratio of the sum of AP to BG (AP/BG); $EEA_{P:N}$ is the ratio of the sum of AP to NAG + LAP (AP/ (NAG + LAP)); The terms $B_{N:C}$, $B_{N:P}$, $B_{P:C}$, and $B_{P:N}$ represent the ratios of MBN to MBC (MBN/MBC), MBN to MBP (MBN/MBP), MBP to MBC (MBP/MBC), and MBP to MBN (MBP / MBN), respectively. Similarly, $L_{N:C}$, $L_{N:P}$, $L_{P:C}$, and $L_{P:N}$ indicate the ratios of TN to SOC (TN / SOC), TN to TP (TN/TP), TP to SOC (TP/SOC), and TP to TN (TP/TN), respectively.

To assess the drivers of microbial nutrient use efficiency, seven soil parameters and five climatic and vegetation parameters were selected as independent variables. The soil parameters included soil depth, pH, SOC, silt and clay content, base saturation (Bsat), and cation exchange capacity (CEC). The climatic and vegetation parameters comprised MAT, MAP, relative humidity, potential evapotranspiration

(PET), and normalized difference vegetation index (NDVI). Latitude-longitude coordinates, SOC, MAT, and MAP were extracted directly from the original articles. For studies where any of these parameters were not reported, missing data were supplemented from global gridded data based on sampling coordinates. Other parameters, including soil pH, silt and clay contents, Bsat, CEC, relative humidity, PET, and NDVI, were directly obtained from global gridded data using sampling coordinates, as these were rarely reported in source articles. Specifically, NDVI values represent annual averages corresponding to the sampling time. These parameters were used to establish relationships between environmental variables and microbial NUE or PUE. To further derive global gridded maps of microbial NUE and PUE through these quantitative relationships, we acquired the same six soil parameters and five climate/vegetation parameters from global databases. A comprehensive list of global data sources for these parameters was provided in Supplementary Table 1. Outliers were identified using the interquartile range (IQR) method, where values falling below 1.5 times the IQR of the first quartile or above 1.5 times the IQR of the third quartile were flagged. These outliers were rigorously reviewed and either corrected or excluded if deemed erroneous, thereby ensuring the robustness of the dataset.

To identify the optimal model for predicting microbial NUE and PUE, we evaluated predictive models, including four linear regression models (multiple linear regression, stepwise regression, elastic net, and least angle regression) and four nonlinear models (Cubist, bagged tree, boosted tree, and Random Forest) (Supplementary Table 1). Model performance was assessed using a repeated Monte Carlo cross-validation approach with 100 resampling iterations implemented in the R package "caret" (v. 6.0-86)[54,55]. For each iteration, the dataset was randomly partitioned into 80% training and 20% validation data. Performance metrics (coefficient of determination ($R^2$) and root mean square error (RMSE)) represent the mean values across all 100 validation iterations, ensuring robust model assessment and mitigating overfitting[56]. The Random Forest model was selected for its superior performance and subsequently used to predict microbial NUE and PUE at a global scale using gridded datasets of the thirteen predictors. Global 1 km resolution maps of microbial NUE and PUE were generated by applying the trained Random Forest model to global covariate layers. The corresponding relative uncertainty of prediction was calculated as follows: for each grid cell, the standard deviation (SD) of predictions was derived from the distribution of outputs across all 500 decision trees in the Random Forest ensemble[57]. This SD represents the range of possible predictions based on model internal variability. The relative uncertainty was then computed by dividing the SD by the global mean microbial NUE and PUE, respectively (Supplementary Figs. 1, 2). This normalization expresses uncertainty as a percentage of the global average, enabling intuitive interpretation and cross-region comparison of uncertainty magnitude. To address multicollinearity among predictors, the variable inflation factor (VIF) was calculated for all variables. Variables with VIFs exceeding 5 were excluded, ensuring that the remaining variables had VIFs below 5[58]. The relative importance of each predictor was quantified using the mean decrease in accuracy (% IncMSE)[37]. Statistical significance was assessed using one-sided permutation tests (500 permutations), with $p$-values representing the probability of obtaining an importance score greater than expected by chance under the null distribution. $p$-values were uncorrected for multiple comparisons. Differences in microbial NUE and PUE across biomes were assessed using one-way ANOVA, followed by Tukey's post-hoc test ($p < 0.05$). All statistical analyses and visualizations were performed using ArcGIS 10.5 and the R environment (v. 4.0.2)[59].

## Reporting summary
Further information on research design is available in the Nature Portfolio Reporting Summary linked to this article.

## Data availability

The data generated in this study have been deposited in a public repository and are available via figshare at https://doi.org/10.6084/m9.figshare.30391051. Global gridded dataset of SOC, TN, pH, silt, clay, CEC, and Bsat, https://soilgrids.org[51]; Global gridded dataset of TP, http://global-change.bnu.edu.cm/research/soilw[52]; Global gridded dataset of MBC, MBN, MBP, https://doi.org/10.5281/zenodo.695062453; Global cropland data, http://www.earthstat.org/cropland-pasture-area-2000/[60]; Global wetland data, https://data.giss.nasa.gov/landuse/wetland.html[61]; MAT and MAP data, www.chelsa-climate.org[62]; PET data, https://doi.org/10.6084/m9.figshare.7504448.v5[63]; Humidity data, https://doi.org/10.3354/cr021001[64]; NDVI data, https://doi.org/10.5067/MEaSUREs/VIP/VIP30.004[65]. All the references used in this study are presented in Supplementary Data 1.

## Code availability

The necessary codes for generating the figures in this study are publicly available at https://doi.org/10.6084/m9.figshare.30391051.

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

## Acknowledgements

We extend our sincere gratitude to the authors of the 213 primary studies (listed in Supplementary Data 1) whose data were essential for this study. This work was financially supported by the National Natural Science Foundation of China (32330071 (H.W.), 32371658 (D.G.), 42301052 (Z.L.)), the Natural Science Foundation of Jilin Province, China (20240101068JC (Z.L.)), and the "Kezhen-Bingwei" Young Talents (2024000204 (D.G.)). Y.C. also acknowledges financial support from the Research Fellowship of the Alexander von Humboldt Foundation. Y.K. thanks the RUDN University Strategic Academic Leadership Program.

## Author contributions

Initial conceptualization: D.G., Y.C., M.D-B., and R.L.S. Methodology: D.G., Y.C., Y.K., and S.W. Investigation: D.G., Y.C., S.W., L.S., M.D-B., and Z.L. Framing, conceptualization and Interpretation: D.G., Y.C., Y.K., M.D-B., J.P., and M.C.R. Visualization: D.G., Y.C., S.W., and L.S. Funding acquisition: D.G., Y.K., and Y.C. Project administration: D.G. Supervision: Y.C. Writing – original draft: D.G. and Y.C. Writing – review & editing: D.G., Y.K., M.D-B., J.P., D.L.M., R.L.S., X.X., L.S., H.W., L.K., X.F., X.D., S.M., Z.L., S.W., F.H., M.C.R., Y.C.

## Funding

## Competing interests

The authors declare no competing interests.
