## [Transparent Peer Review file · Nature Communications]

Global patterns and drivers of soil microbial nitrogen and phosphorus use efficiency

Corresponding Author: Dr Yongxing Cui

Version 0:

Reviewer comments:

Reviewer #1

(Remarks to the Author)

Nutrient use efficiency for soil microbes, including NUE and PUE, is important trait depicting the functions and adaptation of the microbes to environments. However, we still have limited understanding of the global pattern and drivers of the NUE and PUE for soil microbes. Here, the authors collected global dataset of NUE and PUE and explored how the two microbial traits vary along the latitudes. After reading the MS, there are several major issues that should be carefully considered: 1. There is insufficient description of the basic conceptions such as NUE and PUE. For example, NUE and PUE in the Abstract is described as the allocation of assimilated nutrients between growth and mineralization. This is very confusing for understanding of the NUE and PUE. Although the Introduction have given some information showing what means of high NUE and PUE (L71-72), the definition of the PUE and NUE is still unclear to readers. 2. There is unclear about the mechanisms for global patterns of NUE and PUE. For example, stoichiometric homeostasis of the microbes has been used to explain the latitude trend of NUE, but how stoichiometric homeostasis is related to the latitude trend of PUE is unclear. Readers also unclear about why NUE and CUE are decoupled, and the argument of the underlying mechanism is confusing. 3. The difference of NUE and PUE between forests and grasslands is one of the key topics of this study. However, readers have no idea on why the researchers need to explore the difference of the two vegetations. 4. There needs a section in the Introduction to brief introduce the information about the previous studies on NUE and PUE, e.g., the variation of NUE and PUE, key factors influencing the two microbial traits.

More specific issues are listed below:

L48-49 Unclear about the definition of the nutrient use efficiency.

L115-117 Give an example to show whether the NUE and PUE increase or decrease under the case of sufficient C supply.

L117-119 Similarly, are the NUE and PUE of the forests higher or lower than those of the grasslands.

L126-127 Information should be given on how the authors avoid or settle the issues in the collection the NUE and PUE, including different methods used to measure these microbial traits, difficulty in determining the microbial traits such as N or P demanding.

L140-150 These are the main content of the Conclusion, and should not been given here. Please also note that the results of NUE and PUE are different from those in L120-122.

L158-159 in middle to high SOC, SOC has a contrasting effect on NUE with that of case of low SOC, and why such different effects of SOC?, its implication? Importantly, the authors seem to stress the similar findings of this study as those of previous studies. Then, what is the novel finding this study.

L167-170 Readers wonder why carbon-rich conditions will cause high NUE and PUE? And how this is related to stoichiometric homeostasis? Generally, carbon is rich in high SOC soils, which can provide more energy to soil microbes, and more nutrients can be integrated into microbial biomass, and hence less nutrients are released to soils. Therefore, how the stoichiometric homeostasis affect NUE and PUE needs to be articulated.

L176-177 The main topic in this paragraph is on the effects of SOC on nutrient use efficiency, but not the effect of vegetation.

L183-184 Generally, fungi and bacteria have different optimized temperature to growth, not the 30 oC. If I remember correctly, optimized temperature for fungi is about 25 oC and for bacterial is about 35 oC. Then, the authors know what I want to question your argument here. That is under 30 oC, NUE of the fungi is not always increase with temperature.

L186-190 Does it mean the Fig. 3i for the over 80% moisture? However, after the moisture of 80%, there is obvious increase of NUE, different from the authors' description here.

L196-198 For Fig. 3i and Fig. 4i, the authors stated that NUE and PUE initially decreased and then stabilized. However, the

two figures show a bit different trends of NUE and PUE with PET from your statement here. Please carefully check!
L200-202 Why soil microbes tend to catabolic rather than anabolic processes under carbon limitation? Are there any physiological mechanisms for this phenomenon?

L225-227 The authors argue that plant roots and soil fungi compete greatly on soil nutrients in high latitudes. However, this could not be the case because low temperature in this area limits the activities of the roots and the fungi, which is consistent with the authors arguments in L231-232 showing low nutrient demanding of the microbes in this condition. Therefore, it is more likely that microbes especially ECM fungi in this condition secret more enzyme for decomposing SOM for nutrient acquisition, and hence relative lower nutrients used for growth, that is lower NUE.

L242-244 NUE and CUE have opposite trends with latitudes, and this can lead to negative correlation of them, and this is also the coupling not decoupling. However, the results show decoupled NUE and CUE. Therefore, L242-244 is important for clarify why they are decoupled. However, L242-244 is very confusing and readers are very hard to understand the underlying mechanism. For example, which factor reduce plant-microbe competition, and how this can lead to the decoupling of NUE and CUE.

L248-252 The reasons for why PUE show no trend with latitude are too general and not convinced. For example, generally, one factor causes the decrease of PUE with increasing latitude and the other factor causes the increase of PUE with increasing latitude, and hence PUE could show no relationship with latitude because of the offset effect of the two factors. However, there is no such discussion of the mechanisms for lacking of latitude pattern of PUE.

L261 Forest root litter also has more lignin-rich compounds than grass roots?

Fig. 3: The authors should use exact values of the variables in x-axis rather than the low to high trend.

Reviewer #2

(Remarks to the Author)

The authors of Gao et al compiled a large dataset of soil extracellular enzyme activities and used it to explore controls on microbial nutrient use efficiency and make global maps of these traits using gridded data. Some of the authors have published extensively on global soil enzyme compilations before, but not exactly using this framework and with a focus on nutrient use efficiency. As such this seems a new and valuable contribution as it explores macro scale patterns and gives insight into some of the most important environmental factors that control nutrient use efficiency (through an enzyme lens). The discussion of how global trends in carbon vs nutrient use efficiency differ were interesting, as were the disparate patterns observed with nitrogen vs phosphorus efficiency along latitudinal gradients.

The paper is generally well-written and I was overall convinced by the analyses and interpretations, but I did have some specific areas of concern that are outlined below. I also feel that more details are needed in certain areas (especially with the input data) to make this paper truly reproducible and transparent.

Line 95 – ‘Third, stoichiometric models offer a promising...’ this is a bit awkward because the First and Second reasons in the list lead with problems, while the Third leads with a possible solution (although problems with it are later discussed). Can you put the problem up-front so the text reads more consistently?

Line 120 – Although the introduction is written well, I don’t think the rationale for the hypotheses is clearly explained. Why would higher soil C and lower metabolic rates increase nutrient use efficiency (or vice-versa in the tropics)? Perhaps you can add a bit more to explain these predictions.

Lines 142-150 – I find it strange to describe the results in detail before showing the data. One sentence on the key findings seems ok (e.g., Line 141), but in my opinion the rest could be cut.

Figure 1 –It appears your data coverage in the true tropics is limited. Do you think it's fair to make global maps that include the tropics using models developed with so few representative data points? Are you sure there isn't more out there, perhaps some of the papers in this meta-analysis of tropical soil enzymes by Waring et al (<https://link.springer.com/article/10.1007/s10533-013-9849-x>)?

Figure 2, panels (b) and (d) – these R2 values are for the 20% validation dataset, yes? Is it the average of the five cross-folds, or did you pick the fold that had highest accuracy?

Line 225 – Since what you found did not agree with your hypotheses, I might rephrase this to: “In high-latitude biomes, our hypothesis of higher nutrient use efficiency was not supported. This may be due to intense plant-microbe competition for limited N...” or something similar.

Line 237 – Here when discussing the tropics, you also invoke ‘intense plant-microbe competition,’ but in this case it explains high NUE. Please address why at high latitude this leads to low efficiency, but near the poles it goes the opposite way. Also, see comment above about very sparse data in tropical regions.

Figure 5 – There are some very dark areas in the PUE figure in North America, what are your thoughts on what is driving that?

Line 392 – Is it acceptable to use variables to predict/upscale a process when those same variables were used to generate estimates for that process in the first place? Meaning SOC, TN, TP. I feel this needs justification.

Line 396 – Which parameters were extracted from the original articles? I'm confused because later you say MAT and MAP are from WorldClim. So the other three? When was NDVI from, the same time as enzymes were measured, or an annual average?

Table S1 – Are all of these data sources cited in the references, along with providing the links? Links can go down/become obsolete, so doi's are best. I am also a bit confused because you mention SoilGrids in the text, but the links in S1 mostly are to the HWSD (Harmonized World Soil Database), which is not the same thing. Please clarify.

Line 412 – 5-fold seems like a low number for cross-validation. How stable were your results across the folds? Why not use more, like 100-fold? Did you have evidence the data partition to train vs validate did not influence the results?

Lines 420-421 – If you used a random forest model, with gridded datasets and global coverage, to predict NUE and PUE, where does the kriging (e.g., spatial interpolation) come in? I am confused why this is needed, please explain.

Figs S1 & S2 – The uncertainty would be nice to include in the main text, but I'm not clear on the method used for this. Where did you get 'mean absolute error' for each pixel, and why normalize to the global mean (Lines 427-429)?

Data: In the spreadsheets, it is not indicated where each row came from, e.g., which published study. Please add an extra column with the doi or full citation of the original data source (manuscript). This would increase transparency and reproducibility of your work. I also wonder why the spreadsheet doesn't include latitude and longitude for each observation, plus all of the variables mentioned in Lines 336-339. Why are only some things included?

Also, I am not sure how this is usually handled in meta-analysis, but I see that you did not cite all 265 articles that provided data for this paper. That seems fair, but is there a way to give the authors of these papers 'credit' for the use of their data? As above, maybe at least including this in the data spreadsheets would be sufficient.

Reviewer #3

(Remarks to the Author)

This is an intriguing paper, which uses extracellular enzyme activities to assess nutrient imbalances and from that to estimate microbial nitrogen and phosphorus use efficiencies (NUE and PUE). It produces global patterns of these activities. I have two problems with the work though. First, the datasets are limited in their spatial scope. They are concentrated in North America, Europe, and China and are quite limited in the boreal and tropical regions. Africa and South America have almost no measurements. Thus, it is hard to call this a truly global dataset or to accept the conclusions about global patterns. Second and more problematic is that the paper argues that NUE is highest in the tropics and lowest in the boreal and tundra regions. But the tropics are typically P limited and N is dominated by NO₃. In contrast, boreal regions are N limited and are dominated by NH₄. Tundra ecosystems are about the most N limited on earth and plants rely on amino acids for their N nutrition. N is rapidly immobilized. Thus, the direct biogeochemical studies suggest that microbial N assimilation dominates in high latitude environments whereas N mineralization dominates in tropical environments. That is the opposite pattern than that reported by this paper. To me, that suggests that there must be something amiss with the exoenzyme based estimate of NUE—I trust the biogeochemical patterns that have been developed by many direct studies of N cycling. The authors struggle to rationalize why NUE should be high in the tropics and low in high latitudes, and they fail. I think they've shown that the method is fundamentally flawed.

99: "These models can leverage widely available ecological data, and thus produced estimates of microbial NUE and PUE are comparable across large spatial scales." This sentence makes no sense as written.

148: "However, are driven by different environmental controls across biomes, highlighting the complex interplay between climate, 149 vegetation, and soil properties in regulating microbial nutrient dynamics." Something is missing in this sentence—"however are driven"? What are driven?

170: "In contrast, C-limited environments may shift microbial metabolism from catabolic toward anabolic processes, reducing NUE and PUE5" Is this backward? Wouldn't anabolic processes be more effective at assimilating N and P, whereas catabolic processes would lead to mineralization.

213: "Globally, the highest mean NUE values occurred in tropical/subtropical regions (0.66-0.69), followed by temperate regions (0.59-0.62), with the lowest values observed in boreal 215 regions (0.53)." But in boreal and tundra regions, microbial immobilization dominates the N cycle, whereas in tropical environments, mineralization tends to dominate. So this pattern seems to contradict what direct N cycling measurements tend to show. How do you reconcile these different measurement approaches?

236: "In tropical ecosystems, soils are typically N and P co-limited due to rapid mineralization and intense plant-microbe competition" But tropical systems are typically N rich—they mineralize N. They are generally considered to be P limited. Rapid mineralization indicates that soils are not limited.

238: "To sustain growth under strong N competition, microorganisms prioritize N retention, reflecting in higher NUE," But "prioritizing N retention" means limiting N mineralization and you just noted that tropical soils mineralize N! Rather, boreal and tundra systems show strong immobilization and N limitation.

242: "However, reduce plant-microbe competition and efficient microbial N recycling may lead to N saturation, lowering microbial NUE in these environments" But tundra environments are among the most N limited on earth and plant microbe competition is intense. Added N is immediately immobilized. The direct measurements indicate that microbial NUE must be very high. .

299: "Microbial NUE exhibited a clear latitudinal gradient, peaking in tropical/subtropical regions and declining in boreal areas, likely reflecting decreased C inputs with latitude that constrain N allocation to microbial biomass." But high latitude soils may have lower C inputs but they are generally more C rich. Decomposition is slow and N mineralization is limited. Microbes are N limited suggesting that NUE should be high.

Version 1:

Reviewer comments:

Reviewer #1

(Remarks to the Author)

I'm glad to see a great improvement of this version relative to the previous one. Here, I still have some minor comments that should be taken into account in the revision.

L159-160 It is difficult to assume intense plant-microbe competition for boreal or tundra ecosystems.

L230-231 low SOC soil means more labile C in the soils?

L256-257 Do you mean the fig. 3C?

L275-277 Please note that in the dry conditions, i.e., very near MAP = 0, NUE decrease with a little increase of MAP. This seems contrasting with what you described here, Fig. 3B.

L396-397: "more heterogeneous cycling of P relative to N". Please explain this more clearly.

The y-axis in Fig. 4 should be "Predicted PUE"

Reviewer #2

(Remarks to the Author)

I re-reviewed the paper by Gao et al. titled 'Global patterns and factors driving nutrient use efficiency of soil microorganisms.' I appreciate the revisions and like the expanded 'uncertainties' section but have a few remaining concerns.

I still feel there are reproducibility issues with this dataset. The figshare link provided (<https://figshare.com/s/6741ba3d1cf7f64d64ad>) has 'derived' datasets used to generate figures and the R code used to create them, but where is the 'raw' enzyme data? I am expecting one row per observation compiled by the authors, with enzyme values pulled from published literature and ideally each row has a doi or citation for where that enzyme data came from. Apologies if I am missing it, but I don't see this key file. This seems critical for full traceability so that others could be able to verify or build upon the work.

I am also unclear where the stoichiometric ratios for microbial biomass and soil organic matter came from, critical for the ecoenzymatic model (equations 3-6). Did every paper with enzyme data provide these values? If not, where did they come from? Are these values provided in the file in figshare?

Regarding the definition of nutrient use efficiency (Abstract line 49), should it be 'invested in' instead of 'released via' mineralization? As Reviewer 3 points out, we know nitrogen mineralization rates (flux) are highest in the tropics, so if this number is a mineralized N quantity it must be larger at low latitude, which should lead to smaller (not larger) NUE values since it is the denominator. But this is the opposite of what you found. However, if we're talking about 'investment in' nitrogen mineralization enzymes, this definition would make more sense. To address the valid points raised by Reviewer 3, I think it helps to repeatedly mention throughout this paper that the enzymatic approach is about metabolic investment/strategy, not fluxes or process rates per se. This would be good to highlight more clearly, that nutrient use efficiency is not the same as nutrient limitation (and what we learn from studying the former that's different from the latter).

Specific:

Line 52: seem to be missing some words in this sentence, should perhaps be 'we predicted *the controls on* microbial nitrogen *use* efficiency...' or similar. Not sensible as written.

Line 60: could you add a phrase for why NUE is lower at high latitude, same as you have for the PUE phrase?

Lines 303-306: I do not understand how this paradox works. If higher C:N and C:P in the decomposing substrate forces more investment in nutrient acquiring enzymes, shouldn't NUE and PUE values be lower? Can you clarify why not?

Reviewer #3

(Remarks to the Author)

I think that the authors have addressed my concerns adequately. They have substantially rewritten the manuscript. I would recommend accepting the paper.

Version 2:

Reviewer comments:

Reviewer #2

(Remarks to the Author)

I appreciate the second round of revisions, namely the addition of the 'raw' enzyme file to Figshare plus additional manuscript text to clarify the approach and context for the work. All of my comments and concerns have been addressed and I am excited to see this study be published.

Response to Reviewer #1:

Comment 1: *Nutrient use efficiency for soil microbes, including NUE and PUE, is important trait depicting the functions and adaptation of the microbes to environments. However, we still have limited understanding of the global pattern and drivers of the NUE and PUE for soil microbes. Here, the authors collected global dataset of NUE and PUE and explored how the two microbial traits vary along the latitudes. After reading the MS, there are several major issues that should be carefully considered: 1. There is insufficient description of the basic conceptions such as NUE and PUE. For example, NUE and PUE in the Abstract is described as the allocation of assimilated nutrients between growth and mineralization. This is very confusing for understanding of the NUE and PUE. Although the Introduction have given some information showing what means of high NUE and PUE (L71-72), the definition of the PUE and NUE is still unclear to readers.*

Response 1: Thank you for your nice suggestion. In the revised manuscript, we have explicitly redefined microbial NUE and PUE in both the Abstract and the first paragraph of the Introduction.

“Nutrient use efficiency of soil microorganisms, the proportion of assimilated nutrients allocated into biosynthesis rather than released via mineralization, regulates key soil processes such as carbon cycles.” (Lines 48-51)

“Soil microorganisms break down high-molecular-weight organic compounds using coenzymes and assimilate the resulting low-molecular-weight organic N and P compounds^{3,4}. Microbial NUE and PUE represent the proportion of these assimilated small-molecule organic N or P compounds allocated into biosynthesis (primarily growth) versus mineralized release, reflecting the metabolic trade-off between nutrient assimilation and nutrient mineralization⁵⁻⁷. Higher microbial NUE and PUE thus reflect an efficient cycling between using

available N and P for biomass production versus releasing them back into the environment^{5,7}.” (Lines 74-86)

Comment 2: *There is unclear about the mechanisms for global patterns of NUE and PUE. For example, stoichiometric homeostasis of the microbes has been used to explain the latitude trend of NUE, but how stoichiometric homeostasis is related to the latitude trend of PUE is unclear. Readers also unclear about why NUE and CUE are decoupled, and the argument of the underlying mechanism is confusing.*

Response 2: In the revised manuscript, we have now clarified that the lack of a latitudinal pattern in microbial PUE is due to the dominant role of local-scale factors (e.g., soil mineralogy, P retention capacity, and land use) that override climatic gradients. We also explicitly linked these local-scale factors to the principle of stoichiometric homeostasis: microbes strive to maintain a stable internal C:N:P ratio, and local soil conditions exert a stronger influence on their ability to achieve this balance for P compared to the broader climatic gradients that influence N cycling. In addition, the slower and more heterogeneous nature of P cycling compared to N further weakens any broad-scale trend. These points are now explicitly discussed in the revised manuscript (Lines 394-411).

We have revised the manuscript to clarify the mechanisms underlying the divergence between NUE and CUE. Specifically, we now emphasize that CUE is primarily constrained by temperature-dependent respiration, whereas NUE is more strongly influenced by microbial nutrient acquisition strategies and competitive dynamics. We proposed that these contrasting influences are mediated by the principle of stoichiometric homeostasis: microbes adjust their investment in respiration (thereby in CUE) and nutrient acquisition (thereby in NUE) to maintain internal C:N:P stoichiometric balance under different environmental conditions. This understanding, which integrates the influence

of both temperature and nutrient availability, represents a potential critical mechanism governing both CUE and NUE. Specifically, in tropical regions, high respiratory costs reduce CUE, while strong microbial-plant competition for N maintains high NUE. In high-latitude systems, low temperature favors high CUE by reducing respiration, but enzymatic investment in decomposition and pulsed N release lower NUE. This contrast in drivers explains their divergent latitudinal patterns. A more detailed mechanistic explanation has been added in the revised manuscript (Lines 358-387). We believe that these changes fully address the original confusion.

“This lack of apparent global pattern likely attributes to three main reasons: (i) the pervasive P limitation across biomes, (ii) the inherently slower and more heterogeneous cycling of P compared to N, which represses microbial capacity to regulate P use, and (iii) dominant local-scale controls such as soil age, mineralogy, and land use^{19,42}. Moreover, specific latitudinal drivers may offset each other, for instance, pronounced geochemical P retention in tropical soils counterbalances temperature-driven stimulation of mineralization, while lower geochemical high-latitude P limitation is partially offset by slower organic matter turnover.” (Lines 394-411)

“The observed decline in microbial NUE with latitudes (Fig. 5a) contrasts with the well-documented latitudinal increase in microbial CUE^{19,35,36}, revealing a microbial trade-off between these two traits across climatic gradients. In tropical ecosystems, soils experience strong P limitation while maintaining tight N cycling^{19,37}. Despite rapid N mineralization rates, microorganisms prioritize N retention through efficient immobilization-recycling pathways to counter intense plant competition and leaching losses^{19,38-40}, resulting in higher NUE (Fig. 5a). Moreover, elevated respiratory costs at high temperatures reduce C retention, thereby lowering microbial CUE. In boreal and tundra ecosystems, the scenario is reversed. Cold temperatures slow

organic matter decomposition, favoring C immobilization and thus higher microbial CUE³⁵. However, microbial NUE derived from enzymatic stoichiometry declines because energy allocation is prioritized toward the production of extracellular enzymes for decomposition of recalcitrant organic matter rather than microbial biomass synthesis^{26,34}, diverting assimilated N away from growth. Additionally, pulsed N release during thaw events can create transient rises in N availability exceeding microbial demand⁴¹. Thus, while CUE is largely governed by temperature constraints on metabolic efficiency, NUE could be influenced more strongly by microbial nutrient acquisition strategies and competitive dynamics.” (Lines 358-387)

Comment 3: *The difference of NUE and PUE between forests and grasslands is one of the key topics of this study. However, readers have no idea on why the researchers need to explore the difference of the two vegetations.*

Response 3: To better contextualize the significance of comparing forest and grassland in microbial NUE and PUE, we have further focused the discussion on the implications of these differences for ecosystem responses to global change drivers (e.g., N deposition, warming, and elevated CO₂). Specifically, we now emphasized how inherently higher NUE and PUE in forests, driven by stoichiometric imbalances, may increase their resilience to nutrient losses under N deposition or warming, potentially buffering them against declines in primary productivity. In contrast, lower NUE and PUE in grasslands could increase vulnerability to leaching and reduce long-term C sequestration, potentially exacerbating climate change. These clarifications suggest that quantifying these biome-specific strategies is vital for predicting nutrient retention, C storage, and greenhouse gas feedbacks under future climate scenarios.

“This adaptation is crucial for resilience to global change: Under projected

increases in atmospheric CO₂ (which can further raise plant tissue C:N ratios) coupled with warming (altering decomposition kinetics), the high nutrient use efficiency of forest ecosystems may buffer against nutrient losses. Grasslands, with their lower baseline NUE and PUE driven by lower C:nutrient ratios, may be more susceptible to nutrient leaching losses under warming-induced mineralization pulses, potentially reducing long-term C sequestration potential. Consequently, quantifying the specific microbial N and P use strategies between forests and grasslands provides a vital mechanistic basis for modeling how these major biomes will regulate nutrient retention, C storage, and greenhouse gas feedback under future climate scenarios.” (Lines 450-460)

Comment 4: *There needs a section in the Introduction to brief introduce the information about the previous studies on NUE and PUE, e.g., the variation of NUE and PUE, key factors influencing the two microbial traits.*

Response 4: Thank you for this constructive suggestion. We have added a section in the introduction to explicitly summarize current knowledge on variations in NUE and PUE and their potential drivers.

“Microbial nutrient use efficiency varies across terrestrial ecosystems, primarily due to widespread environmental and resource constraints in microbial metabolisms^{12,15}. Previous studies have revealed that NUE and PUE depend on biome type, climate, and soil properties, with higher efficiencies in forests than grasslands or croplands^{5,15,16}. The availability and stoichiometric balance of nutrients in soils are likely the major factors influencing microbial NUE and PUE^{5,8,17}, as microorganisms are able to adjust metabolic strategies to maintain stoichiometric homeostasis. Competition between soil microorganisms and plants as well as between microorganisms for N and P could further modulate microbial nutrient use strategies and

efficiencies, with both groups dynamically reallocating limited resources in response to changing environmental conditions^{4,18,19}. Consequently, anthropogenic nutrient inputs such as atmospheric N deposition may alter the magnitudes of microbial NUE and PUE^{5,8,20}. Specifically, organic C availability plays a critical role in microbial activity and growth^{12,15,21}, both soil organic carbon (SOC) and plant-derived C inputs could thus be key drivers of spatial patterns of microbial NUE and PUE^{16,22,23}. For example, incubation experiments suggest that sufficient C source supply (e.g., via labile C input) elevates microbial NUE and PUE by alleviating energy constraints^{5,9,24}, which explains high microbial NUE and PUE in forests, as forest soils typically have higher SOC content and C:N ratios than grasslands²⁵. In contrast, climate conditions such as temperature and precipitation are also important regulators of microbial activity and vary across biomes, which may lead to biome-specific difference in microbial NUE and PUE. However, it is unclear whether high microbial NUE and PUE in forests are primarily driven by C availability or by climate, and which factors act as the key drivers of their global patterns.” (Lines 124-158)

Comment 5: L48-49 Unclear about the definition of the nutrient use efficiency.

Response 5: In the revised manuscript, we have explicitly redefined microbial NUE and PUE in the Abstract.

“Nutrient use efficiency of soil microorganisms, the proportion of assimilated nutrients allocated into biosynthesis rather than released via mineralization, regulates key soil processes such as carbon cycles.” (Lines 48-51)

Comment 6: L115-117 Give an example to show whether the NUE and PUE increase or decrease under the case of sufficient C supply.

Response 6: Thank you for your suggestion. We have added the following

explicit example in the revised Introduction.

“For example, incubation experiments suggest that sufficient C source supply (e.g., via labile C input) elevates microbial NUE and PUE by alleviating energy constraints^{5,9,24},” (Lines 147-149)

Comment 7: L117-119 *Similarly, are the NUE and PUE of the forests higher or lower than those of the grasslands.*

Response 7: Yes, both microbial NUE and PUE are higher in forests than in grasslands. We have clarified this point in the revised manuscript.

“For example, incubation experiments suggest that sufficient C source supply (e.g., via labile C input) elevates microbial NUE and PUE by alleviating energy constraints^{5,9,24}, which explains high microbial NUE and PUE in forests, as forest soils typically have higher SOC content and C:N ratios than grasslands²⁵.” (Lines 147-151)

Comment 8: L126-127 *Information should be given on how the authors avoid or settle the issues in the collection the NUE and PUE, including different methods used to measure these microbial traits, difficulty in determining the microbial traits such as N or P demanding.*

Response 8: Thank you for this nice suggestion. In response, we have now explicitly stated that only studies using consistent and widely accepted methods, chloroform fumigation-extraction for microbial biomass and fluorometric assays for enzyme activities, were included in our meta-analysis. This approach minimizes variability arising from methodological differences and enhances the reliability of derived NUE and PUE estimates.

“To ensure consistency and comparability across studies, we only included datasets where microbial biomass was determined using the chloroform fumigation-extraction method, and extracellular enzyme activities were measured based on fluorometric substrate assays, which are widely adopted standard protocols in soil microbial ecology.” (Lines 176-180)

Comment 9: *L140-150 These are the main content of the Conclusion, and should not been given here. Please also note that the results of NUE and PUE are different from those in L120-122.*

Response 9: We agree with you and have incorporated them into the Conclusion to improve the flow and conciseness of the manuscript.

Comment 10: *L158-159 in middle to high SOC, SOC has a contrasting effect on NUE with that of case of low SOC, and why such different effects of SOC? its implication? Importantly, the authors seem to stress the similar findings of this study as those of previous studies. Then, what is the novel finding this study.*

Response 10: Thank you for this insightful comment. The differential effect of SOC on microbial NUE across low versus medium-to-high SOC ranges is a key point that merits further clarification and discussion. In SOC-poor soils, increasing SOC alleviates energy limitation and facilitates greater microbial investment in nutrient acquisition, thereby enhancing NUE. Our findings suggest that in these soils, even relatively small increases in SOC can lead to substantial improvements in NUE. This could have significant implications for strategies aimed at restoring degraded soils. However, in soils with medium to high SOC levels, nutrient availability or stoichiometric constraints may become increasingly dominant, causing NUE to stabilize despite further increases in SOC. This implies that in these systems, simply adding more C may not be sufficient to further enhance nutrient retention or C sequestration,

and therefore other interventions (e.g., nutrient additions, management of plant litter) may be necessary. This shift in the limiting factors, from energy to nutrients, helps explain the contrasting relationships across SOC gradients.

While previous studies have established the general importance of SOC in microbial nutrient use efficiency, our work provides the first global-scale empirical evidence of the dual role of SOC in regulating microbial NUE and PUE, acting as an efficiency enhancer in C-limited soils; but exhibiting constrained effects under high SOC due to emerging nutrient limitation. This finding challenges the assumption that simply increasing SOC will always lead to greater nutrient retention and C sequestration. We have revised the manuscript to more clearly emphasize these novel insights and have removed statements that overemphasized consistency with previous studies.

“Partial regression analysis indicated that both efficiencies increased markedly with rising SOC levels in soils with low initial SOC content ($p < 0.05$; Figs. 3 and 4), likely because added C alleviates energy limitation and stimulates microbial investment in nutrient acquisition via enzymes. However, this positive trend plateaued under medium to high SOC levels (Figs. 3 and 4), suggesting a shift from energy to nutrient limitation. In these environments, microbial nutrient use efficiency may be constrained by the availability of N or P rather than C, highlighting the stoichiometric constraints on microbial metabolism. In low-SOC soils, labile substrate addition supplies readily available C and energy to support microbial metabolism, enabling more efficient assimilation of N and P into biomass²⁴. This reduces nutrient losses via excretion or mineralization, thereby increasing microbial NUE and PUE⁵. SOC could also favor the production of extracellular enzymes such as β -glucosidase (BG), N-acetyl-glucosaminidase (NAG), and acid phosphatase (AP), all of which are essential for nutrient acquisition¹⁴. Under C-rich conditions, soil microorganisms prioritize nutrient retention to maintain cellular

stoichiometric homeostasis^{5,20,26}. The surplus energy from SOC allows greater investment in assimilatory processes relative to dissimilatory pathways, thereby raising NUE and PUE by increasing biomass growth. These SOC-driven mechanisms are further supported by positive correlations between microbial NUE and PUE and the normalized difference vegetation index (NDVI) ($p < 0.05$; Figs. 3 and 4), as higher NDVI reflects increased plant-derived C inputs^{22,27}. Further, with increases in C availability, limited nutrient availability may become restrict in NUE and PUE, reflecting a transition from energy to nutrient limitation.” (Lines 222-254).

“These findings underscore the dosage-effect of SOC in regulating microbial nutrient use efficiency.” (Lines 256-257)

Comment 11: *L167-170 Readers wonder why carbon-rich conditions will cause high NUE and PUE? And how this is related to stoichiometric homeostasis? Generally, carbon is rich in high SOC soils, which can provide more energy to soil microbes, and more nutrients can be integrated into microbial biomass, and hence less nutrients are released to soils. Therefore, how the stoichiometric homeostasis affect NUE and PUE needs to be articulated.*

Response 11: As you pointed out, abundant SOC provides ample energy, enabling microorganisms to allocate more resources toward assimilatory processes (i.e., incorporating nutrients into biomass) rather than dissimilatory pathways (e.g., respiration or nutrient mineralization). This shift in metabolic strategy enhances the efficiency of nutrient incorporation into microbial biomass, thereby directly increasing NUE and PUE. Moreover, this process is fundamentally linked to stoichiometric homeostasis: microorganisms strive to maintain stable internal C:N:P ratios. Under C-rich conditions, the demand for N and P increases to balance the excess C. By increasing NUE and PUE,

microbes improve their retention of N and P, reducing nutrient loss and supporting the maintenance of cellular stoichiometric balance. In the revised manuscript, we have modified the explanation to more explicitly articulate this mechanism.

“Under C-rich conditions, soil microorganisms prioritize nutrient retention to maintain cellular stoichiometric homeostasis^{5,20,26}. The surplus energy from SOC allows greater investment in assimilatory processes relative to dissimilatory pathways, thereby raising NUE and PUE by increasing biomass growth.” (Lines 241-246)

Comment 12: L176-177 *The main topic in this paragraph is on the effects of SOC on nutrient use efficiency, but not the effect of vegetation.*

Response 12: To further avoid any misunderstanding, we have rephrased this sentence to explicitly tie vegetation’s role to SOC dynamics.

“These SOC-driven mechanisms are further supported by positive correlations between microbial NUE and PUE and the normalized difference vegetation index (NDVI) ($p < 0.05$; Figs. 3 and 4), as higher NDVI reflects increased plant-derived C inputs^{22,27}.” (Lines 246-249)

Comment 13: L183-184 *Generally, fungi and bacteria have different optimized temperature to growth, not the 30 °C. If I remember correctly, optimized temperature for fungi is about 25 °C and for bacterial is about 35 °C. Then, the authors know what I want to question your argument here. That is under 30 °C, NUE of the fungi is not always increase with temperature.*

Response 13: Yes, we agree that using a single temperature value (such as 30 °C) overlooks important physiological differences between these microbial

groups. We have now clearly stated the temperature optima for fungal (~25 °C) and bacterial (~35 °C) growth, and more accurately describes the divergent effects of warming on NUE and PUE across microbial groups in the revised manuscript.

“Below thermal optima (~ 25 °C for fungi, ~ 35 °C for bacteria), increasing temperature generally increases microbial growth rates^{28,29}. This raises greater allocation of N and P toward anabolic processes, thereby elevating microbial NUE and PUE.” (Lines 269-273)

Comment 14: *L186-190 Does it mean the Fig. 3i for the over 80% moisture? However, after the moisture of 80%, there is obvious increase of NUE, different from the authors' description here.*

Response 14: In fact, the mentioned moisture threshold (> 80%) was derived from prior studies, not our dataset. For avoiding any confusing, we have revised the wording in the revised manuscript to prevent potential misinterpretation by readers.

“Soil moisture regulates microbial metabolism bidirectionally: under dry conditions, rising moisture typically increases microbial NUE and PUE by alleviating water limitation for microorganisms and nutrient diffusion constraints^{15,30}. In contrast, when moisture exceeds certain levels that cause oxygen depletion, hypoxic stress may reduce microbial growth and nutrient use efficiency^{15,31}.” (Lines 273-282)

Comment 15: *L196-198 For Fig. 3i and Fig. 4i, the authors stated that NUE and PUE initially decreased and then stabilized. However, the two figures show a bit different trends of NUE and PUE with PET from your statement here. Please carefully check!*

Response 15: Yes, the patterns of NUE and PUE along PET are slightly different. For avoiding the oversimplified description for Fig. 3i and 4i patterns, we have rephrased this sentence in the revised manuscript.

“Specifically, NUE showed an initial transient increase followed by rapid decline, secondary rise, and subsequent stabilization at high PET levels. In contrast, PUE increased initially before declining and stabilizing at elevated PET.” (Lines 288-291)

Comment 16: *L200-202 Why soil microbes tend to catabolic rather than anabolic processes under carbon limitation? Are there any physiological mechanisms for this phenomenon?*

Response 16: In C-limited conditions, soil microbes face critical energy shortages. To survive, they must prioritize immediate energy generation through catabolic processes (such as respiration) to produce ATP, which is essential for basic cellular maintenance and survival functions. This explanation has now been incorporated into the revised manuscript to explicitly address the physiological basis for the shift toward catabolism under C scarcity.

“The decline phases reflect rising microbial stress under high PET, where increased aridity and temperature typically induce C limitation and potentially N accumulation. Under these PET-induced conditions, microbial communities may shift metabolic priorities from growth to maintenance, reducing nutrient use efficiency. Specifically, microbial NUE decreases as C scarcity forces microorganisms to prioritize catabolic adenosine triphosphate (ATP) generation for cellular maintenance, while substantial N resources are diverted toward the production of C-acquiring coenzymes^{13,15,33}.” (Lines 292-303)

Comment 17: L225-227 *The authors argue that plant roots and soil fungi compete greatly on soil nutrients in high latitudes. However, this could not be the case because low temperature in this area limits the activities of the roots and the fungi, which is consistent with the authors arguments in L231-232 showing low nutrient demanding of the microbes in this condition. Therefore, it is more likely that microbes especially ECM fungi in this condition secrete more enzyme for decomposing SOM for nutrient acquisition, and hence relative lower nutrients used for growth, that is lower NUE.*

Response 17: We agree with your opinions that low temperatures limit the activity of both roots and fungi, making direct nutrient competition less likely. Instead, the observed lower NUE arises primarily because microbes, especially ECM fungi, allocate more assimilated nutrients toward enzyme production for SOM decomposition rather than biomass growth under energy and nutrient constraints. We have revised the discussion accordingly to clarify this mechanism in the revised manuscript.

“Contrary to our hypothesis that high-latitude biomes would have higher microbial NUE, our results revealed lower values in these regions (Fig. 5 and Table 1). Our observation of lower NUE in cold regions, based on the ecoenzymatic stoichiometric approach, indicates that microorganisms allocate a greater portion of assimilated N to enzyme production for organic matter decomposition. This aligns with the concept that microbial NUE is governed not simply by N availability or temperature constraints, but by a combination of competitive dynamics and microbial resource allocation strategies. In nutrient-limited environments, microbial communities (e.g., ectomycorrhizal fungi) prioritize enzymatic investment for decomposing recalcitrant organic matter over biomass production^{26,34}. This C and energy reallocation diverts assimilated N from growth to enzyme synthesis, explaining the observed reduction in NUE. The resulting mobilization of organically bound N may partially alleviate N limitation despite ecosystem-level scarcity. Moreover, the

fungal dominance in high-latitude biomes, characterized by high C:N ratio and lower stoichiometric discrepancies with recalcitrant substrates such as lignin-rich litter, may further reduce microbial N demand and contribute to lower NUE²⁶. Accordingly, the lower NUE in high-latitude biomes reflects microbial strategies to adapt N-poor conditions and are therefore consistent with the general view that these cold ecosystems are primarily N-limited.” (Lines 329-357)

Comment 18: *L242-244 NUE and CUE have opposite trends with latitudes, and this can lead to negative correlation of them, and this is also the coupling not decoupling. However, the results show decoupled NUE and CUE. Therefore, L242-244 is important for clarify why they are decoupled. However, L242-244 is very confusing and readers are very hard to understand the underlying mechanism. For example, which factor reduce plant-microbe competition, and how this can lead to the decoupling of NUE and CUE.*

Response 18: Thank you for these suggestions. In the revised manuscript, we have removed the term 'decoupling' to avoid implying a complete lack of connection between these two traits. Instead, we aim to clarify that several factors influence NUE and CUE more strongly in certain ecosystems compared to others. We emphasize that CUE is primarily constrained by temperature-dependent respiration, whereas NUE is more strongly influenced by microbial nutrient acquisition strategies and competitive dynamics. For example, in tropical regions, high respiratory costs reduce CUE, because microorganisms must expend more energy to maintain basic metabolic functions, thus there is less available energy available for growth and biomass. Although plant and microbial competition for N exists, the actual amount is far less important of an impact than the overall CUE. High microbial-plant competition also exists in reverse in higher latitude biomes, but again, is far less important than temperatures restricting enzymatic investments. A more detailed mechanistic explanation has been added in the revised manuscript.

“The observed decline in microbial NUE with latitudes (Fig. 5a) contrasts with the well-documented latitudinal increase in microbial CUE^{19,35,36}, revealing a microbial trade-off between these two traits across climatic gradients. In tropical ecosystems, soils experience strong P limitation while maintaining tight N cycling^{19,37}. Despite rapid N mineralization rates, microorganisms prioritize N retention through efficient immobilization-recycling pathways to counter intense plant competition and leaching losses^{19,38-40}, resulting in higher NUE (Fig. 5a). Moreover, elevated respiratory costs at high temperatures reduce C retention, thereby lowering microbial CUE. In boreal and tundra ecosystems, the scenario is reversed. Cold temperatures slow organic matter decomposition, favoring C immobilization and thus higher microbial CUE³⁵. However, microbial NUE derived from enzymatic stoichiometry declines because energy allocation is prioritized toward the production of extracellular enzymes for decomposition of recalcitrant organic matter rather than microbial biomass synthesis^{26,34}, diverting assimilated N away from growth. Additionally, pulsed N release during thaw events can create transient rises in N availability exceeding microbial demand⁴¹. Thus, while CUE is largely governed by temperature constraints on metabolic efficiency, NUE could be influenced more strongly by microbial nutrient acquisition strategies and competitive dynamics.” (Lines 358-387)

Comment 19: L248-252 *The reasons for why PUE show no trend with latitude are too general and not convinced. For example, generally, one factor causes the decrease of PUE with increasing latitude and the other factor causes the increase of PUE with increasing latitude, and hence PUE could show no relationship with latitude because of the offset effect of the two factors. However, there is no such discussion of the mechanisms for lacking of latitude pattern of PUE.*

Response 19: We agree that a more explicit discussion of the offsetting mechanisms is needed. In the revised manuscript, we have now included specific examples of counteracting factors: for instance, in tropical regions, intensified geochemical P retention opposes temperature-enhanced mineralization, while in high latitudes, strong P limitation is partially compensated by slower organic matter turnover. These additions clarify how divergent latitudinal drivers can mask a clear PUE pattern.

“This lack of apparent global pattern likely attributes to three main reasons: (i) the pervasive P limitation across biomes, (ii) the inherently slower and more heterogeneous cycling of P compared to N, which represses microbial capacity to regulate P use, and (iii) dominant local-scale controls such as soil age, mineralogy, and land use^{19,42}. Moreover, specific latitudinal drivers may offset each other, for instance, pronounced geochemical P retention in tropical soils counterbalances temperature-driven stimulation of mineralization, while lower geochemical high-latitude P limitation is partially offset by slower organic matter turnover.” (Lines 394-411)

Comment 20: *L261 Forest root litter also has more lignin-rich compounds than grass roots?*

Response 20: Yes, forest root litter typically contains higher lignin content compared to grass roots, which further reinforces the stoichiometric imbalance mechanism we proposed. To address this, we have revised the sentence to explicitly include root-derived litter in our explanation.

“This reflects stronger stoichiometric imbalances in forest ecosystems, where soils typically have higher C:N and C:P ratios due to lignin-rich plant litter (both foliar and root-derived) compared to grassland residues^{45,46}.” (Lines 427-430)

Comment 21: *Fig. 3: The authors should use exact values of the variables in x-axis rather than the low to high trend.*

Response 21: As suggested, we have revised Figs. 3 and 4 to display exact values of the variables on the x-axis instead of the original "low to high" gradient representation. The updated figures have been included in the revised manuscript.

Response to Reviewer #2:

Comment 1: *The authors of Gao et al compiled a large dataset of soil extracellular enzyme activities and used it to explore controls on microbial nutrient use efficiency and make global maps of these traits using gridded data. Some of the authors have published extensively on global soil enzyme compilations before, but not exactly using this framework and with a focus on nutrient use efficiency. As such this seems a new and valuable contribution as it explores macro scale patterns and gives insight into some of the most important environmental factors that control nutrient use efficiency (through an enzyme lens). The discussion of how global trends in carbon vs nutrient use efficiency differ were interesting, as were the disparate patterns observed with nitrogen vs phosphorus efficiency along latitudinal gradients.*

Response 1: We thank the reviewer for their positive evaluation of our work and for recognizing the novelty and value of our global-scale analysis of microbial nutrient use efficiency through an enzymatic stoichiometry framework.

Comment 2: *The paper is generally well-written and I was overall convinced by the analyses and interpretations, but I did have some specific areas of concern that are outlined below. I also feel that more details are needed in certain areas (especially with the input data) to make this paper truly reproducible and transparent.*

Response 2: We thank the reviewer for their positive assessment of our manuscript and for the constructive suggestions. We have carefully addressed all the specific concerns raised, particularly regarding the need for greater detail on input data to enhance reproducibility and transparency. Detailed descriptions of data sources and analysis have been added to the Methods section (Lines 613-625 and 642-666) and the Supplementary

Materials (See Table S1). We believe these revisions have significantly improved the clarity and rigor of the study.

“Latitude-longitude coordinates, soil pH, SOC, silt and clay content, MAT, and MAP were extracted directly from the original articles. For studies where any of these parameters were not reported, missing data were supplemented from global gridded data based on sampling coordinates. Other parameters including Bsat, CEC, relative humidity, PET, and NDVI were directly obtained from global gridded data using sampling coordinates, as these were rarely reported in source articles. Specifically, NDVI values represent annual averages corresponding to the sampling time. These parameters were used to establish relationships between environmental variables and microbial NUE or PUE. To further derive global gridded maps of microbial NUE and PUE through these quantitative relationships, we acquired the same six soil parameters and five climate/vegetation parameters from global databases. A comprehensive list of global data sources for these parameters was provided in Table S1.” (Lines 613-625)

“Model performance was assessed using a repeated Monte Carlo cross-validation approach with 100 resampling iterations implemented in the R package "caret" (v. 6.0-86)^{53,54}. For each iteration, the dataset was randomly partitioned into 80% training and 20% validation data. Performance metrics (coefficient of determination (R^2) and root mean square error (RMSE)) represent the mean values across all 100 validation iterations, ensuring robust model assessment and mitigating overfitting⁵⁵. The Random Forest model was selected for its superior performance and subsequently used to predict microbial NUE and PUE at a global scale using gridded datasets of the thirteen predictors. Global 1 km resolution maps of microbial NUE and PUE were generated by applying the trained Random Forest model to global

covariate layers. The corresponding relative uncertainty of prediction was calculated as follows: for each grid cell, the standard deviation (SD) of predictions was derived from the distribution of outputs across all 500 decision trees in the Random Forest ensemble⁵⁶. This SD represents the range of possible predictions based on model internal variability. The relative uncertainty was then computed by dividing this SD by the global mean value of microbial NUE and PUE, respectively (Figs. S1, S2). This normalization expresses uncertainty as a percentage of the global average, enabling intuitive interpretation and cross-region comparison of uncertainty magnitude.”
(Lines 642-666)

Comment 3: *Line 95 – ‘Third, stoichiometric models offer a promising...’ this is a bit awkward because the First and Second reasons in the list lead with problems, while the Third leads with a possible solution (although problems with it are later discussed). Can you put the problem up-front so the text reads more consistently?*

Response 3: We agree that maintaining a consistent structure can improve readability and logical flow. As suggested, we have revised the third point to begin by highlighting the challenge associated with stoichiometric models, specifically the difficulty in obtaining necessary parameters for large-scale applications.

“Third, while stoichiometric models enable large-scale estimation of microbial NUE and PUE^{13,14}, their implementation requires multiple difficult-to-obtain parameters (e.g., microbial biomass, microbial activities, and soil nutrient supply), which has critically constrained their applications in large scales.”
(Lines 108-112)

Comment 4: *Line 120 – Although the introduction is written well, I don’t think*

the rationale for the hypotheses is clearly explained. Why would higher soil C and lower metabolic rates increase nutrient use efficiency (or vice-versa in the tropics)? Perhaps you can add a bit more to explain these predictions.

Response 4: We have revised the hypotheses to clarify the underlying mechanisms. Specifically, in high-latitude regions, N limitation and slow decomposition promote microbial nutrient conservation, increasing NUE. In contrast, tropical systems exhibit rapid mineralization, reducing nutrient retention and NUE. We also distinctively addressed PUE, expecting highest efficiency under P-limited tropical conditions. The hypothesis section has been thoroughly revised in the revised manuscript.

“Here, we hypothesized higher microbial NUE in boreal/tundra ecosystems due to N limitation and intense plant-microbial competition under slow-decomposition regimes, whereas NUE could lower in the tropics, where rapid mineralization and leaching reduce microbial retention. Microbial PUE could follow an inverse pattern with the higher value in P-limited tropical soils but lower value in higher latitudes where P limitation is less pronounced.” (Lines 159-164)

Comment 5: *Lines 142-150 – I find it strange to describe the results in detail before showing the data. One sentence on the key findings seems ok (e.g., Line 141), but in my opinion the rest could be cut.*

Response 5: Thank you for your suggestions, we have now removed this paragraph in the revised manuscript.

Comment 6: *Figure 1 –It appears your data coverage in the true tropics is limited. Do you think it’s fair to make global maps that include the tropics using models developed with so few representative data points? Are you sure there isn’t more out there, perhaps some of the papers in this meta-analysis of*

tropical soil enzymes by Waring et al
(<https://link.springer.com/article/10.1007/s10533-013-9849-x>)?

Response 6: We acknowledge that the relatively sparse representation of tropical sites introduces greater uncertainty for tropical regions. We have explicitly addressed this concern in the Uncertainties section of the revised manuscript, noting that spatial heterogeneity in sampling intensity, especially in the tropics, affects model reliability.

Regarding the meta-analysis by Waring et al., we had carefully examined the studies included, and the case is that 9 of the 19 tropical datasets were already incorporated in our synthesis. The remaining 10 were excluded due to lacking measurements of one or more essential enzymes (BG, NAG, or AP) required for calculating NUE and PUE stoichiometrically (see Table 1 below). We agree that future research should prioritize collecting more data from tropical ecosystems to improve the accuracy and robustness of global models of microbial nutrient dynamics when more observations from tropical ecosystems are available.

“Second, despite the global scope of our dataset, spatial heterogeneity in sampling intensity introduces uncertainty, particularly in tropical and boreal ecosystems where site density is lowest (Fig. 1). Therefore, prioritizing studies in these underrepresented regions would be valuable for strengthening the robustness of future studies.” (Lines 488-492)

Table 1 Literature inclusion/exclusion status of tropical studies from Waring et al. (2014) with rationales for non-Incorporation.

Number	References	Included?	Critical missing enzymes
1	Acosta-Martinez et al., 2007	No	NAG and LAP
2	Badiane et al., 2001	No	NAG, LAP, and AP
3	Caldwell et al., 1999	No	NAG, LAP, and AP
4	Chacon et al., 2009	No	BG, NAG, and LAP
5	Cusack et al., 2011	Yes	—

6	Dinesh et al., 2004	No	NAG, LAP, and AP
7	King et al., 2008	Yes	—
8	Olander et al., 2000	No	BG, NAG, and LAP
9	Sandoval-Perez et al., 2009	Yes	—
10	Sjogersten et al., 2011	No	LAP and AP
11	Sotomayor-Ramirez et al., 2009	No	NAG, LAP, and AP
12	Turner, 2010	Yes	—
13	Turner et al., 2010	Yes	—
14	Ushio et al., 2010	Yes	—
15	Verchot et al., 2005	Yes	—
16	Waldrop et al., 2000	No	NAG and LAP
17	Weintraub et al., 2012	Yes	—
18	Wick et al., 2000	Yes	—
19	Yavitt et al., 2004	No	BG, NAG, and LAP

Reference:

Waring, B.G., Weintraub, S.R. and Sinsabaugh, R.L. (2014). Ecoenzymatic stoichiometry of microbial nutrient acquisition in tropical soils. *Biogeochemistry*, 117(1): 101-113.

Comment 7: *Figure 2, panels (b) and (d) – these R² values are for the 20% validation dataset, yes? Is it the average of the five cross-folds, or did you pick the fold that had highest accuracy?*

Response 7: Yes, that is correct. The R^2 values shown in Fig. 2b and 2d represent the model performance on the 20% validation datasets. Importantly, these values are the average R^2 across all 100 cross-validation folds, ensuring a robust and generalized estimate of model accuracy rather than selecting any single high-accuracy fold. We have clarified this point in the Methods section of the revised manuscript.

“Performance metrics (coefficient of determination (R^2) and root mean square error (RMSE)) represent the mean values across all 100 validation iterations, ensuring robust model assessment and mitigating overfitting⁵⁵.” (Lines 646-648)

Comment 8: *Line 225 – Since what you found did not agree with your hypotheses, I might rephrase this to: “In high-latitude biomes, our hypothesis of higher nutrient use efficiency was not supported. This may be due to intense plant-microbe competition for limited N...” or something similar.*

Response 8: Thank you for your specific suggestions. We have rephrased this sentence in the revised manuscript as suggested.

“Contrary to our hypothesis that high-latitude biomes would have higher microbial NUE, our results revealed lower values in these regions (Fig. 5 and Table 1).” (Lines 329-331)

Comment 9: *Line 237 – Here when discussing the tropics, you also invoke ‘intense plant-microbe competition,’ but in this case it explains high NUE. Please address why at high latitude this leads to low efficiency, but near the poles it goes the opposite way. Also, see comment above about very sparse data in tropical regions.*

Response 9: Thank you for the careful review regarding the NUE patterns across different latitudes. In the revised manuscript, we have provided a more detailed explanation of the distinct mechanisms driving microbial NUE in different latitudinal zones. Specifically, we clarified that in high-latitude regions, the dominance of fungi and their investment of assimilated N into enzyme production for decomposing recalcitrant organic matter—rather than into growth—results in lower NUE.

“Contrary to our hypothesis that high-latitude biomes would have higher microbial NUE, our results revealed lower values in these regions (Fig. 5 and Table 1). Our observation of lower NUE in cold regions, based on the coenzymatic stoichiometric approach, indicates that microorganisms allocate a greater portion of assimilated N to enzyme production for organic matter

decomposition. This aligns with the concept that microbial NUE is governed not simply by N availability or temperature constraints, but by a combination of competitive dynamics and microbial resource allocation strategies. In nutrient-limited environments, microbial communities (e.g., ectomycorrhizal fungi) prioritize enzymatic investment for decomposing recalcitrant organic matter over biomass production^{26,34}. This C and energy reallocation diverts assimilated N from growth to enzyme synthesis, explaining the observed reduction in NUE. The resulting mobilization of organically bound N may partially alleviate N limitation despite ecosystem-level scarcity. Moreover, the fungal dominance in high-latitude biomes, characterized by high C:N ratio and lower stoichiometric discrepancies with recalcitrant substrates such as lignin-rich litter, may further reduce microbial N demand and contribute to lower NUE²⁶. Accordingly, the lower NUE in high-latitude biomes reflects microbial strategies to adapt N-poor conditions and are therefore consistent with the general view that these cold ecosystems are primarily N-limited.” (Lines 329-357)

Comment 10: *Figure 5 – There are some very dark areas in the PUE figure in North America, what are your thoughts on what is driving that?*

Response 10: The very dark areas (indicating high microbial PUE) in North America are likely driven by a combination of high soil C:P ratio and specific geochemical conditions, which lead to potentially high P limitation in soil microorganisms. In fact, we recent study suggest there are high microbial P limitation in this area (Cui et al. 2025). For the specific mechanisms, (1) High soil C:P ratio: These regions are characterized by cold climates, coniferous vegetation that produces low-quality (high C:P ratio) litter, and acidic soils. This creates a strong demand for P relative to C, selecting for microbial communities that efficiently retain and recycle P (high PUE) rather than releasing it. (2) Soil mineralogy: These acidic soils are often rich in iron and aluminum (oxyhydr) oxides, which have a high capacity to adsorb and retain soluble phosphate, making it unavailable to microbes and plants. To

counteract this strong geochemical retention, microbes may enhance their internal P use efficiency to cope with the limited available P pool. We have incorporated these into Discussion and clarified that local-scale factors can create regional hotspots of high PUE.

“However, we found some regional hotspots of microbial PUE, such as the high-PUE hotspots in North America (Fig. 5b), which could be explained by the combination of strong P limitation from coniferous litter and the strong P sorption via metal oxides, which promotes highly efficient microbial P conservation strategies⁴³.” (Lines 411-417)

Comment 11: *Line 392 – Is it acceptable to use variables to predict/upscale a process when those same variables were used to generate estimates for that process in the first place? Meaning SOC, TN, TP. I feel this needs justification.*

Response 11: We fully acknowledge this concern. We have now removed both TN and TP from the set of predictor variables in the model (Line 609), as they were direct components in the NUE and PUE calculation formulae, respectively, which would create a direct mathematical interdependence.

However, we retained SOC as a predictor based on a critical conceptual distinction. The SOC values used as predictors represent the overall pool of organic C in the soil, a fundamental property of the soil environment that serves as a key indicator of general energy availability and habitat conditions. Its role in the model is theorized to influence microbial metabolic efficiency and community dynamics from the perspective of the broader environmental context.

This is conceptually distinct from the role of C in the calculation of NUE and PUE. While SOC is involved in the calculation of NUE/PUE denominators (TN, TP are fractions of SOC), the C flows involved in the efficiency calculations

are specifically related to microbial assimilation and growth efficiencies from the utilized substrate. The size of the total SOC pool influences the metabolic environment and resource availability but is not arithmetically incorporated into the efficiency calculations themselves. Therefore, using the total SOC pool as a predictor is theoretically justified to test hypotheses about how broader soil nutrient and energy status govern microbial metabolic strategies, without creating a direct mathematical circularity.

Furthermore, to ensure the robustness of our model and mitigate any concerns of overfitting or collinearity, we performed multicollinearity diagnostics and excluded all variables with a variance inflation factor (VIF) greater than 5 (Lines 672-675). A VIF score indicates the degree to which a predictor variable is explained by other predictors in the model, and a threshold of 5 is commonly used to identify and remove variables with excessive multicollinearity. These steps substantially reduce the potential circularity and multicollinearity, but we acknowledge that some degree of residual correlation between SOC and NUE and PUE may still exist. This could potentially lead to a slight overestimation of the importance of SOC as a predictor. We have added a brief discussion of this potential limitation in the Uncertainties section of the revised manuscript (Lines 500-504).

“To address multicollinearity among predictors, the variable inflation factor (VIF) was calculated for all variables. Variables with a VIF exceeding 5 were excluded, ensuring that the remaining variables had a VIF below 5⁵⁷.” (Lines 672-675)

“Lastly, a limitation arises from using SOC as a predictor for global NUE and PUE estimates, since SOC was also used to estimate NUE and PUE in the stoichiometry models. Despite VIF screening (threshold ≤ 5) to reduce multicollinearity, residual correlations may persist, slightly overestimating the

predictive importance of SOC.” (Lines 500-504)

Comment 12: *Line 396 – Which parameters were extracted from the original articles? I’m confused because later you say MAT and MAP are from WorldClim. So the other three? When was NDVI from, the same time as enzymes were measured, or an annual average?*

Response 12: The parameters initially attempted to be extracted from the original articles were latitude-longitude coordinates, soil pH, SOC, silt and clay content, MAT, and MAP. For any missing data, these were supplemented from global databases. Parameters, including Bsat, CEC, relative humidity, PET, and NDVI, were consistently obtained from global databases, as they were rarely reported in the original articles. The NDVI values represent annual averages corresponding to the sampling year. Detailed explanations regarding the parameter acquisition methods have now been provided in the revised Methods section.

“Latitude-longitude coordinates, soil pH, SOC, silt and clay content, MAT, and MAP were extracted directly from the original articles. For studies where any of these parameters were not reported, missing data were supplemented from global gridded data based on sampling coordinates. Other parameters including Bsat, CEC, relative humidity, PET, and NDVI were directly obtained from global gridded data using sampling coordinates, as these were rarely reported in source articles. Specifically, NDVI values represent annual averages corresponding to the sampling time. These parameters were used to establish relationships between environmental variables and microbial NUE or PUE. To further derive global gridded maps of microbial NUE and PUE through these quantitative relationships, we acquired the same six soil parameters and five climate/vegetation parameters from global databases. A comprehensive list of global data sources for these parameters was provided

in Table S1.” (Lines 613-625)

Comment 13: *Table S1 – Are all of these data sources cited in the references, along with providing the links? Links can go down/become obsolete, so doi's are best. I am also a bit confused because you mention SoilGrids in the text, but the links in S1 mostly are to the HWSD (Harmonized World Soil Database), which is not the same thing. Please clarify.*

Response 13: We are sorry for the inconsistency between Table S1 and the manuscript text. We have now added DOIs and formal references for all data sources in the revised Table S1 (see Table S1).

Comment 14: *Line 412 – 5-fold seems like a low number for cross-validation. How stable were your results across the folds? Why not use more, like 100-fold? Did you have evidence the data partition to train vs validate did not influence the results?*

Response 14: Actually, in our analysis, we employed a repeated random sub-sampling validation (Monte Carlo cross-validation) approach with 100 iterations, using an 80/20 train/validation split for each iteration. This approach was chosen specifically to enhance the stability and reliability of the performance estimates across different random partitions of the data, as recommended for robust model assessment and to mitigate the risk of overfitting. The mean R^2 and RMSE reported are the averages across these 100 validation sets. We have now corrected them in the Methods section accordingly.

“Model performance was assessed using a repeated Monte Carlo cross-validation approach with 100 resampling iterations implemented in the R package "caret" (v. 6.0-86)^{53,54}. For each iteration, the dataset was randomly partitioned into 80% training and 20% validation data. Performance metrics

(coefficient of determination (R^2) and root mean square error (RMSE)) represent the mean values across all 100 validation iterations, ensuring robust model assessment and mitigating overfitting⁵⁵.” (Lines 642-648)

Comment 15: *Lines 420-421 – If you used a random forest mode, with gridded datasets and global coverage, to predict NUE and PUE, where does the kriging (e.g., spatial interpolation) come in? I am confused why this is needed, please explain.*

Response 15: Thank you for pointing this out. Upon careful review, we discovered an unintentional error regarding the kriging in the original manuscript. In fact, kriging was not used in our spatial prediction process. The global maps of microbial NUE and PUE were generated solely by applying the trained Random Forest model to gridded, globally continuous environmental datasets. To avoid any misunderstanding, we have now revised corresponding contents in the revised manuscript.

“The Random Forest model was selected for its superior performance and subsequently used to predict microbial NUE and PUE at a global scale using gridded datasets of the thirteen predictors. Global 1 km resolution maps of microbial NUE and PUE were generated by applying the trained Random Forest model to global covariate layers.” (Lines 653-657)

Comment 16: *Figs S1 & S2 – The uncertainty would be nice to include in the main text, but I’m not clear on the method used for this. Where did you get ‘mean absolute error’ for each pixel, and why normalize to the global mean (Lines 427-429)?*

Response 16: Thank you for your nice suggestions. After careful discussion and evaluation, we decided to keep the uncertainty maps (Figs. S1 and S2) in the Supplemental Information, and taken global distribution maps of NUE and

PUE as the primary focus in the main figures to maximize the clarity and impact of our core findings for a broad readership.

We are sorry for the lack of clarity in our original description regarding the relative uncertainty calculation. The uncertainty for each pixel was derived internally from the Random Forest ensemble, calculated as the standard deviation (SD) of predictions across all 500 decision trees. This SD captures the model's internal variability for that location. We normalized this SD by the global mean NUE and PUE to express uncertainty as a relative measure (i.e., a percentage of the global average). This allows for an intuitive and scale-independent comparison of prediction reliability across regions with different absolute values. We have thoroughly revised the Methods section to clarify these steps and the rationale behind them in the revised manuscript.

“The corresponding relative uncertainty of prediction was calculated as follows: for each grid cell, the standard deviation (SD) of predictions was derived from the distribution of outputs across all 500 decision trees in the Random Forest ensemble⁵⁶. This SD represents the range of possible predictions based on model internal variability. The relative uncertainty was then computed by dividing this SD by the global mean value of microbial NUE and PUE, respectively (Figs. S1, S2). This normalization expresses uncertainty as a percentage of the global average, enabling intuitive interpretation and cross-region comparison of uncertainty magnitude.” (Lines 657-666)

Comment 17: *Data: In the spreadsheets, it is not indicated where each row came from, e.g., which published study. Please add an extra column with the doi or full citation of the original data source (manuscript). This would increase transparency and reproducibility of your work. I also wonder why the spreadsheet doesn't include latitude and longitude for each observation, plus all of the variables mentioned in Lines 336-339. Why are only some things*

included?

Response 17: We have included a complete list of all 265 source articles in the Supplementary information (See Table S3 in the Supplementary information). The previously submitted dataset only included the core data for Figure 5, which explains the absence of geographic coordinates. Regarding the variables mentioned in lines 336-339, these are intermediate variables used in the calculation of NUE and PUE but are not included in the plotting datasets. In the revised manuscript, we have uploaded all datasets used in figures, along with the associated R code, and added hyperlinks to these files in the Data Availability (<https://figshare.com/s/6741ba3d1cf7f64d64ad>) and Code Availability (<https://figshare.com/s/6741ba3d1cf7f64d64ad>) sections. This ensures full traceability from raw, compiled data to the results presented in the manuscript.

Comment 18: *Also, I am not sure how this is usually handled in meta-analysis, but I see that you did not cite all 265 articles that provided data for this paper. That seems fair, but is there a way to give the authors of these papers' 'credit' for the use of their data? As above, maybe at least including this in the data spreadsheets would be sufficient.*

Response 18: Thank you for this valuable suggestion. We have included a complete list of all 265 source articles in the Supplementary information (See Table S3 in the Supplementary information). In addition, we agree with that it is crucial to properly acknowledge and credit the authors whose primary data made this meta-analysis possible. We have added one sentence in the Acknowledgements section of the revised manuscript to explicitly express our gratitude to the authors of these primary studies.

“We extend our sincere gratitude to the authors of the 265 primary studies (listed in Table S3) whose data were essential for this study.” (Lines 994-995)

Response to Reviewer #3:

Comment 1: *This is an intriguing paper, which uses extracellular enzyme activities to assess nutrient imbalances and from that to estimate microbial nitrogen and phosphorus use efficiencies (NUE and PUE). It produces global patterns of these activities. I have two problems with the work though. First, the datasets are limited in their spatial scope. They are concentrated in North America, Europe, and China and are quite limited in the boreal and tropical regions. Africa and South America have almost no measurements. Thus, it is hard to call this a truly global dataset or to accept the conclusions about global patterns. Second and more problematic is that the paper argues that NUE is highest in the tropics and lowest in the boreal and tundra regions. But the tropics are typically P limited and N is dominated by NO₃. In contrast, boreal regions are N limited and are dominated by NH₄. Tundra ecosystems are about the most N limited on earth and plants rely on amino acids for their N nutrition. N is rapidly immobilized. Thus, the direct biogeochemical studies suggest that microbial N assimilation dominates in high latitude environments whereas N mineralization dominates in tropical environments. That is the opposite pattern than that reported by this paper. To me, that suggests that there must be something amiss with the exoenzyme based estimate of NUE—I trust the biogeochemical patterns that have been developed by many direct studies of N cycling. The authors struggle to rationalize why NUE should be high in the tropics and low in high latitudes, and they fail. I think they've shown that the method is fundamentally flawed.*

Response 1: Thank you for the insightful review. Yes, there is considerable limitation in the spatial distribution of our observations, but we have tried our best to collect the accessible data currently and have used Random Forest modelling linking the factors controlling NUE and PUE to their global distribution. Nonetheless, as we all know, this is a common challenge in

global-scale ecological synthesis based on aggregated data from published studies. This specific limitation can be addressed only when more observations from the boreal and tropical regions becomes available. To fully disclose this limitation, we have added a more explicit discussion regarding this limitation in the Uncertainties section of the revised manuscript, and suggested that future research should prioritize researches in these critical geographical areas (Lines 488-492).

We agree with your opinion regarding the apparent discrepancy between our observed NUE patterns and those suggested by established biogeochemical studies of N cycling. Our results, at first glance, seem counterintuitive, given the well-documented N limitation and strong microbial N immobilization in high-latitude ecosystems. In fact, our results provide a complementary perspective on microbial nutrient dynamics by capturing a distinct facet of microbial metabolic strategy: the allocation of assimilated nutrients between growth and the production of extracellular enzymes (Zhang et al., 2019). While direct measurements of N cycling reflect the net outcome of immobilization vs. mineralization, our results based on the stoichiometry approach provides insights into the metabolic cost associated with nutrient acquisition. Specifically, our finding of lower NUE in high-latitude biomes suggests that microorganisms in these environments invest a greater proportion of their assimilated N into producing the enzymes to decompose organic matter (particularly ectomycorrhizal fungi), even though the overall effect is strong N immobilization. This represents an upfront investment in infrastructure to access a limited resource, which increases with the degree the resource is limiting such as in N-limited high latitude biomes. Accordingly, the lower NUE in high-latitude biomes reflects microbial strategies to cope with N-poor conditions and therefore does not contradict the general view that these cold ecosystems are primarily N-limited.

In addition, our recent study indicates strong microbial N limitation in tropical

regions, which is primarily caused by inherent P limitation in tropical soils (Cui et al., 2025). In tropical ecosystems, soils experience strong P limitation while maintaining tight N cycling. Despite rapid N mineralization rates, microorganisms prioritize N retention through efficient immobilization-recycling pathways (instead of releasing ecoenzymes) to counter intense plant competition and leaching losses (Averill et al., 2018; Kuzyakov et al., 2013; Thomas et al., 2015), which is reflected in higher NUE.

Moreover, a recent study comparing two commonly used methods (the ecoenzymatic stoichiometric approach, as employed in our study, and the ^{18}O -based isotopic approach) found that these methods yielded diametrically opposite latitudinal trends in microbial nutrient use efficiency (Sun et al., 2024). In fact, these methods capture different microbial processes, extracellular investment strategies versus intracellular assimilation efficiency, which may explain contrasting ecological patterns. Specifically, the ^{18}O method primarily reflects the short-term intracellular metabolic partitioning of organic molecules between catabolic (respiration) and anabolic (biomass synthesis) processes, offering a snapshot of assimilation efficiency under the prevailing substrate and environmental conditions (Geyer et al., 2019; Zhang et al., 2019; Schimel et al., 2022). In contrast, the ecoenzymatic stoichiometric approach emphasizes the microbial response to elemental stoichiometric imbalances between the microbial biomass and its external substrates, focusing on the enzymatic investment strategy to acquire limiting resources from the environment (Mooshammer et al., 2014; Sinsabaugh et al., 2016).

A recent study has confirmed a significant positive relationship between microbial NUE measured using the ^{18}O method and mean annual temperature (Yang et al., 2025). This finding provides further support for the reliability of our results, which show lower NUE values at higher latitudes based on estimates of the ecoenzymatic stoichiometric approach.

Therefore, this interpretation should differ from traditional views of microbial NUE, and we have made substantial revisions to the manuscript to better explain this distinction and provide a more robust mechanistic framework for our findings. Key revisions in the revised manuscript are summarized and listed below:

(1) Clarifying the definition of NUE: We have explicitly redefined microbial NUE in both the Abstract (Lines 48-51) and the Introduction (Lines 74-86) to emphasize its connection to metabolic allocation.

(2) Strengthening the mechanistic explanation: We have added a more detailed discussion of the factors driving NUE patterns in different latitudinal zones, emphasizing the role of ectomycorrhizal fungi in high-latitude ecosystems and P limitation-driven microbial N use processes in the tropics (Lines 334-357 and 365-369).

(3) Addressing methodological differences: We have explicitly discussed how different methods (ecoenzymatic stoichiometry vs. isotopic tracers) capture distinct aspects of microbial NUE, potentially leading to divergent geographical patterns (Lines 472-487).

Finally, we also want to emphasize that while isotopic techniques provide valuable insights into NUE in specific settings, the ecoenzymatic stoichiometry approach remains the only feasible method for large-scale assessments of microbial nutrient use efficiency, particularly for phosphorus (PUE). Therefore, for integrated and comparative analyses of nutrient use efficiency across ecosystems, especially those involving both N and P, the enzyme-based approach offers a practical and unique avenue for exploring how microbial resource use strategies vary across broad spatial scales.

Once again, we acknowledge that considerable uncertainty remains in current methods for measuring microbial NUE. Methodological development might be another priority for advancing the mechanistic understanding of microbial

resource utilization. We have also incorporated this point into the Uncertainties section (Lines 484-487).

We believe that these revisions address the reviewer's concerns and provide a more compelling justification for our approach and results.

References:

- Averill, C. & Waring, B. Nitrogen limitation of decomposition and decay: How can it occur? *Glob. Chang. Biol.* **24**, 1417 – 1427 (2018).
- Cui, Y. et al. Global patterns of nutrient limitation in soil microorganisms. *Proc. Natl. Acad. Sci. U.S. A.* **122**, e2424552122 (2025).
- Geyer, K.M., Dijkstra, P., Sinsabaugh, R. & Frey, S.D. Clarifying the interpretation of carbon use efficiency in soil through methods comparison. *Soil Biol. Biochem.* **128**, 79 – 88 (2019).
- Kuzyakov, Y. & Xu, X. Competition between roots and microorganisms for nitrogen: mechanisms and ecological relevance. *New Phytol.* **198**, 656 – 669 (2013).
- Mooshammer, M. et al. Adjustment of microbial nitrogen use efficiency to carbon:nitrogen imbalances regulates soil nitrogen cycling. *Nat. Commun.* **5**, 3694 (2014).
- Schimmel, J., Weintraub, M.N. & Moorhead, D. Estimating microbial carbon use efficiency in soil: Isotope-based and enzyme-based methods measure fundamentally different aspects of microbial resource use. *Soil Biol. Biochem.* **169**, 108677 (2022).
- Sinsabaugh, R.L. et al. Stoichiometry of microbial carbon use efficiency in soils. *Ecol. Monogr.* **86**, 172 – 189 (2016).
- Sun, L. et al. Interpreting the differences in microbial carbon and nitrogen use efficiencies estimated by ¹⁸O labeling and ecoenzyme stoichiometry. *Geoderma* **444**, 116856 (2024).
- Thomas, R.Q., Brookshire, E.N. & Gerber, S. Nitrogen limitation on land: how can it occur in Earth system models? *Glob. Chang. Biol.* **21**, 1777 – 1793 (2015).
- Yang, J. et al. Intensified aridity hinders soil microbes from improving their nitrogen use efficiency. *Glob. Chang. Biol.* **31**, e70453 (2025).
- Zhang, S., Zheng, Q., Noll, L., Hu, Y. & Wanek, W. Environmental effects on soil microbial nitrogen use efficiency are controlled by allocation of organic nitrogen to microbial growth and regulate gross N mineralization. *Soil. Biol. Biochem.* **135**, 304 – 315 (2019).

“Second, despite the global scope of our dataset, spatial heterogeneity in sampling intensity introduces uncertainty, particularly in tropical and boreal

ecosystems where site density is lowest (Fig. 1). Therefore, prioritizing studies in these underrepresented regions would be valuable for strengthening the robustness of future studies.” (Lines 488-492)

“Nutrient use efficiency of soil microorganisms, the proportion of assimilated nutrients allocated into biosynthesis rather than released via mineralization, regulates key soil processes such as carbon cycles.” (Lines 48-51)

“Soil microorganisms break down high-molecular-weight organic compounds using coenzymes and assimilate the resulting low-molecular-weight organic N and P compounds^{3,4}. Microbial NUE and PUE represent the proportion of these assimilated small-molecule organic N or P compounds allocated into biosynthesis (primarily growth) versus mineralized release, reflecting the metabolic trade-off between nutrient assimilation and nutrient mineralization⁵⁻⁷. Higher microbial NUE and PUE thus reflect an efficient cycling between using available N and P for biomass production versus releasing them back into the environment^{5,7}.” (Lines 74-86)

“Our observation of lower NUE in cold regions, based on the coenzymatic stoichiometric approach, indicates that microorganisms allocate a greater portion of assimilated N to enzyme production for organic matter decomposition. This aligns with the concept that microbial NUE is governed not simply by N availability or temperature constraints, but by a combination of competitive dynamics and microbial resource allocation strategies. In nutrient-limited environments, microbial communities (e.g., ectomycorrhizal fungi) prioritize enzymatic investment for decomposing recalcitrant organic matter over biomass production^{26,34}. This C and energy reallocation diverts assimilated N from growth to enzyme synthesis, explaining the observed reduction in NUE. The resulting mobilization of organically bound N may partially alleviate N limitation despite ecosystem-level scarcity. Moreover, the

fungal dominance in high-latitude biomes, characterized by high C:N ratio and lower stoichiometric discrepancies with recalcitrant substrates such as lignin-rich litter, may further reduce microbial N demand and contribute to lower NUE²⁶. Accordingly, the lower NUE in high-latitude biomes reflects microbial strategies to adapt N-poor conditions and are therefore consistent with the general view that these cold ecosystems are primarily N-limited.” (Lines 334-357)

“In tropical ecosystems, soils experience strong P limitation while maintaining tight N cycling^{19,37}. Despite rapid N mineralization rates, microorganisms prioritize N retention through efficient immobilization-recycling pathways to counter intense plant competition and leaching losses^{19,38-40}, resulting in higher NUE (Fig. 5a).” (Lines 365-369)

“It is also important to note that differences in methodological approaches, such as ecoenzymatic stoichiometry versus isotopic tracers, can lead to divergent interpretations of microbial nutrient use efficiency⁵⁰. For instance, Sun et al.⁵⁰ found that these methods yielded opposing latitudinal trends for microbial NUE, by comparing the two prevalent methods (i.e., ecoenzymatic stoichiometric approach, as employed in our study, and the ¹⁸O-based isotopic approach). In fact, these methods capture different microbial processes, extracellular investment strategies versus intracellular assimilation efficiency, which could reflect contrasting ecological patterns. Compared to ¹⁸O method, the ecoenzymatic stoichiometric approach emphasizes microbial responses to elemental stoichiometric imbalances between microbial biomass and its external substrates, focusing on the enzymatic investment strategy to acquire limiting resources from the environment¹⁴. Therefore, future efforts toward methodological integration and development are essential to reconcile these perspectives and advance the mechanistic understanding of microbial resource use.” (Lines 472-487)

“Therefore, future efforts toward methodological integration and development are essential to reconcile these perspectives and advance the mechanistic understanding of microbial resource use.” (Lines 484-487)

Comment 2: 99: *“These models can leverage widely available ecological data, and thus produced estimates of microbial NUE and PUE are comparable across large spatial scales.” This sentence makes no sense as written.*

Response 2: We agree that this sentence was confusing and poorly constructed. We have removed this sentence entirely from the revised manuscript to improve clarity. (Lines 201-213).

Comment 3: 148: *“However, are driven by different environmental controls across biomes, highlighting the complex interplay between climate, 149 vegetation, and soil properties in regulating microbial nutrient dynamics.” Something is missing in this sentence—“however are driven”? What are driven?*

Response 3: We appreciate you pointing out this grammatical error. The surrounding content was deemed unnecessary following other revisions, and we have therefore removed the related contents in the revised manuscript to improve flow and conciseness.

Comment 4: 170: *“In contrast, C-limited environments may shift microbial metabolism from catabolic toward anabolic processes, reducing NUE and PUE” Is this backward? Wouldn’t anabolic processes be more effective at assimilating N and P, whereas catabolic processes would lead to mineralization.*

Response 4: You are correct that the original phrasing was backward. To avoid any misunderstanding, we have now restructured these sentences in the revised manuscript.

“Under C-rich conditions, soil microorganisms prioritize nutrient retention to maintain cellular stoichiometric homeostasis^{5,20,26}. The surplus energy from SOC allows greater investment in assimilatory processes relative to dissimilatory pathways, thereby raising NUE and PUE by increasing biomass growth. These SOC-driven mechanisms are further supported by positive correlations between microbial NUE and PUE and the normalized difference vegetation index (NDVI) ($p < 0.05$; Figs. 3 and 4), as higher NDVI reflects increased plant-derived C inputs^{22,27}. Further, with increases in C availability, limited nutrient availability may become restrict in NUE and PUE, reflecting a transition from energy to nutrient limitation. These findings underscore the dosage-effect of SOC in regulating microbial nutrient use efficiency.” (Lines 241-257)

Comment 5: 213: *“Globally, the highest mean NUE values occurred in tropical/subtropical regions (0.66-0.69), followed by temperate regions (0.59-0.62), with the lowest values observed in boreal regions (0.53).” But in boreal and tundra regions, microbial immobilization dominates the N cycle, whereas in tropical environments, mineralization tends to dominate. So this pattern seems to contradict what direct N cycling measurements tend to show. How do you reconcile these different measurement approaches?*

Response 5: The observed low NUE in boreal regions, based on the ecoenzymatic approach, reflects the high metabolic cost of N acquisition required to sustain life in N-limited, recalcitrant environments. Microbes must prioritize the allocation of assimilated N toward N-acquiring ecoenzymes, which inherently reduces the efficiency of N allocated to biomass growth (NUE). This high cost occurs even when the net ecosystem outcome is strong immobilization. This nuanced, mechanistic explanation is now clearly articulated in the revised manuscript.

“Our observation of lower NUE in cold regions, based on the ecoenzymatic stoichiometric approach, indicates that microorganisms allocate a greater portion of assimilated N to enzyme production for organic matter decomposition. This aligns with the concept that microbial NUE is governed not simply by N availability or temperature constraints, but by a combination of competitive dynamics and microbial resource allocation strategies. In nutrient-limited environments, microbial communities (e.g., ectomycorrhizal fungi) prioritize enzymatic investment for decomposing recalcitrant organic matter over biomass production^{26,34}. This C and energy reallocation diverts assimilated N from growth to enzyme synthesis, explaining the observed reduction in NUE. The resulting mobilization of organically bound N may partially alleviate N limitation despite ecosystem-level scarcity. Moreover, the fungal dominance in high-latitude biomes, characterized by high C:N ratio and lower stoichiometric discrepancies with recalcitrant substrates such as lignin-rich litter, may further reduce microbial N demand and contribute to lower NUE²⁶. Accordingly, the lower NUE in high-latitude biomes reflects microbial strategies to adapt N-poor conditions and are therefore consistent with the general view that these cold ecosystems are primarily N-limited.” (Lines 334-357)

Comment 6: 236: “In tropical ecosystems, soils are typically N and P co-limited due to rapid mineralization and intense plant-microbe competition” But tropical systems are typically N rich—they mineralize N. They are generally considered to be P limited. Rapid mineralization indicates that soils are not limited.

Response 6: We agree that tropical ecosystems are conventionally considered P-limited. We have revised the text to reflect this primary limitation while maintaining the microbial nuance. This clarification acknowledges that the high microbial NUE results from the necessary strategy of retaining N efficiently under strong competition and P limitation, even though net N fluxes

are rapid.

“In tropical ecosystems, soils experience strong P limitation while maintaining tight N cycling^{19,37}. Despite rapid N mineralization rates, microorganisms prioritize N retention through efficient immobilization-recycling pathways to counter intense plant competition and leaching losses^{19,38-40}, resulting in higher NUE (Fig. 5a).” (Lines 365-369)

Comment 7: 238: *“To sustain growth under strong N competition, microorganisms prioritize N retention, reflecting in higher NUE,” But “prioritizing N retention” means limiting N mineralization and you just noted that tropical soils mineralize N! Rather, boreal and tundra systems show strong immobilization and N limitation.*

Response 7: This comment addresses the apparent conflict between rapid mineralization (a flux) and high microbial retention (a strategy). Our explanation, now detailed in the revised manuscript, reconciles this by demonstrating the CUE-NUE trade-off across latitudes: In the tropics, high temperature leads to high respiratory costs (low CUE), but P limitation and competition drive high N retention (high NUE). In the boreal, cold temperatures favor C retention (high CUE), but the need for enzymatic mining for N diverts assimilated N away from growth (low NUE). Thus, high N mineralization and high microbial N retention are not mutually exclusive; they reflect distinct selective pressures that shape microbial resource allocation strategies, a distinction captured by the ecoenzymatic stoichiometry approach.

“The observed decline in microbial NUE with latitudes (Fig. 5a) contrasts with the well-documented latitudinal increase in microbial CUE^{19,35,36}, revealing a microbial trade-off between these two traits across climatic gradients. In tropical ecosystems, soils experience strong P limitation while maintaining

tight N cycling^{19,37}. Despite rapid N mineralization rates, microorganisms prioritize N retention through efficient immobilization-recycling pathways to counter intense plant competition and leaching losses^{19,38-40}, resulting in higher NUE (Fig. 5a). Moreover, elevated respiratory costs at high temperatures reduce C retention, thereby lowering microbial CUE. In boreal and tundra ecosystems, the scenario is reversed. Cold temperatures slow organic matter decomposition, favoring C immobilization and thus higher microbial CUE³⁵. However, microbial NUE derived from enzymatic stoichiometry declines because energy allocation is prioritized toward the production of extracellular enzymes for decomposition of recalcitrant organic matter rather than microbial biomass synthesis^{26,34}, diverting assimilated N away from growth. Additionally, pulsed N release during thaw events can create transient rises in N availability exceeding microbial demand⁴¹. Thus, while CUE is largely governed by temperature constraints on metabolic efficiency, NUE could be influenced more strongly by microbial nutrient acquisition strategies and competitive dynamics.” (Lines 358-387)

Comment 8: 242: *“However, reduce plant-microbe competition and efficient microbial N recycling may lead to N saturation, lowering microbial NUE in these environments” But tundra environments are among the most N limited on earth and plant microbe competition is intense. Added N is immediately immobilized. The direct measurements indicate that microbial NUE must be very high.*

Response 8: We appreciate the reviewer's comment, which is very useful for reconciling our findings with the well-known fact that tundra ecosystems are highly N-limited and competition is intense. The lower NUE we report in cold regions, as estimated by the ecoenzymatic stoichiometry approach, reflects the high metabolic cost of nutrient acquisition. Microbes in these environments face recalcitrant organic matter, forcing them to prioritize allocating assimilated N into N-acquiring ecoenzymes (an investment cost) over allocation to biomass growth (which defines our NUE metric). Therefore,

the intense competition and strong N immobilization (the ecosystem outcome) are not contradicted by our low enzyme-based NUE; rather, they are consistent with the high upfront metabolic investment required to "mine" for scarce N. This unique perspective on microbial energy allocation is a central finding of our study and is now clarified in the revised manuscript.

“In boreal and tundra ecosystems, the scenario is reversed. Cold temperatures slow organic matter decomposition, favoring C immobilization and thus higher microbial CUE³⁵. However, microbial NUE derived from enzymatic stoichiometry declines because energy allocation is prioritized toward the production of extracellular enzymes for decomposition of recalcitrant organic matter rather than microbial biomass synthesis^{26,34}, diverting assimilated N away from growth. Additionally, pulsed N release during thaw events can create transient rises in N availability exceeding microbial demand⁴¹. Thus, while CUE is largely governed by temperature constraints on metabolic efficiency, NUE could be influenced more strongly by microbial nutrient acquisition strategies and competitive dynamics.” (Lines 375-387)

Comment 9: 299: *“Microbial NUE exhibited a clear latitudinal gradient, peaking in tropical/subtropical regions and declining in boreal areas, likely reflecting decreased C inputs with latitude that constrain N allocation to microbial biomass.” But high latitude soils may have lower C inputs but they are generally more C rich. Decomposition is slow and N mineralization is limited. Microbes are N limited suggesting that NUE should be high.*

Response 9: We agree that high-latitude soils are generally N-limited, and microorganisms are under strong selective pressure to use N efficiently. However, we clarify that our observed NUE pattern is not primarily driven by C inputs alone but by the microbial allocation strategy required for survival in

that environment.

While microbes are N-limited (suggesting high N capture), the coenzymatic approach reveals that this efficiency comes at a cost. The lower NUE observed reflects the diversion of assimilated N toward enzyme production to access N locked in organic matter. This allocation choice, necessitated by the recalcitrance of the substrate, reduces the efficiency with which N is channeled into microbial growth. We have clarified this mechanism, which links the structural N limitations of the substrate to the physiological investment strategies of the microbes, in the revised discussion.

“Our observation of lower NUE in cold regions, based on the coenzymatic stoichiometric approach, indicates that microorganisms allocate a greater portion of assimilated N to enzyme production for organic matter decomposition. This aligns with the concept that microbial NUE is governed not simply by N availability or temperature constraints, but by a combination of competitive dynamics and microbial resource allocation strategies. In nutrient-limited environments, microbial communities (e.g., ectomycorrhizal fungi) prioritize enzymatic investment for decomposing recalcitrant organic matter over biomass production^{26,34}. This C and energy reallocation diverts assimilated N from growth to enzyme synthesis, explaining the observed reduction in NUE. The resulting mobilization of organically bound N may partially alleviate N limitation despite ecosystem-level scarcity. Moreover, the fungal dominance in high-latitude biomes, characterized by high C:N ratio and lower stoichiometric discrepancies with recalcitrant substrates such as lignin-rich litter, may further reduce microbial N demand and contribute to lower NUE²⁶. Accordingly, the lower NUE in high-latitude biomes reflects microbial strategies to adapt N-poor conditions and are therefore consistent with the general view that these cold ecosystems are primarily N-limited.” (Lines 334-357)

Point-to-point response to reviewers' comments

Response to Reviewer #1:

Comment 1: *I'm glad to see a great improvement of this version relative to the previous one. Here, I still have some minor comments that should be taken into account in the revision.*

Response 1: Thank you, we are very glad to hear that you find our revision is a great improvement over the previous one. We have incorporated your further suggestions into the next revision of the paper.

Comment 2: *L159-160 It is difficult to assume intense plant-microbe competition for boreal or tundra ecosystems.*

Response 2: Thank you for this comment. We have revised the hypothesis to emphasize that low temperatures constrain N availability, which likely drive higher microbial NUE in these ecosystems.

“Here, we hypothesized high microbial NUE in boreal/tundra ecosystems due to widespread N limitation in cold regions, whereas NUE could be low in the tropics, where rapid mineralization and leaching reduce microbial N retention.”

(Lines 138-142)

Comment 3: *L230-231 low SOC soil means more labile C in the soils?*

Response 3: No, soils with low SOC typically contain lower levels of labile C. Our intended point was that under such C-limited conditions, the external addition of labile substrates can supply readily available C and energy, thereby enhancing microbial assimilation of N and P. We have revised the text by specifying “exogenous labile substrate addition” to avoid any misunderstanding.

“In low-SOC soils where labile C pools are typically depleted, exogenous labile substrate addition supplies readily available C and energy to support microbial metabolism, enabling more efficient assimilation of N and P into biomass²⁴.” (Lines 183-186)

Comment 4: L256-257 Do you means the fig. 3C?

Response 4: Yes, we have added Fig. 3c at the end of this sentence in the revised manuscript.

“These findings underscore the dosage-effect of SOC in regulating microbial nutrient use efficiency (Fig. 3c).” (Lines 200-201)

Comment 5: L275-277 Please note that in the dry conditions, i.e., very near $MAP = 0$, NUE decrease with a little increase of MAP. This seems contrasting with what you described here, Fig. 3B.

Response 5: We agree that at the extremely low end of the moisture gradient (MAP close to 0), NUE may initially decline with a slight increase in MAP, as shown in Fig. 3b. To avoid this apparent contradiction and better reflect the observed pattern, we have revised the text to refer specifically to “mild to moderate dry conditions,” thereby excluding the extremely arid range you noted.

“under mild to moderate dry conditions, rising moisture typically increases microbial NUE and PUE by alleviating water limitation for microorganisms and nutrient diffusion constraints^{15,30}.” (Lines 210-212)

Comment 6: L396-397: “more heterogenous cycling of P relative to N” .

Please explain this more clearly.

Response 6: This original phrasing was meant to indicate that P availability and transformation processes show much greater local variability than those of N. This is because P cycling is strongly controlled by spatially heterogeneous factors such as soil mineralogy, metal oxides, and geochemical adsorption. In contrast, N cycling is more consistently shaped by broader climatic and biological processes. To improve clarity, we have revised the manuscript by (1) replacing “heterogeneous” with the term “variable” and (2) adding a sentence clarifying that this high local variability in P cycling can obscure larger-scale environmental gradients.

“(ii) the inherently slower and more variable cycling of P compared to N, which represses microbial capacity to regulate P use” (Lines 298-299)

“This resulting high variability in P cycling from such localized heterogeneity may obscure continental or global-scale environmental gradients⁴⁴, which reflects fundamental differences in how microbial stoichiometric homeostasis operates for P vs. N.” (Lines 311-316)

Comment 7: *The y-axis in Fig. 4 should be “Predicted PUE”*

Response 7: Thank you, and we have now revised the y-axis label of Fig. 4 to “Predicted PUE” in the revised manuscript.

Response to Reviewer #2:

Comment 1: *I re-reviewed the paper by Gao et al. titled ‘Global patterns and factors driving nutrient use efficiency of soil microorganisms.’ I appreciate the revisions and like the expanded ‘uncertainties’ section but have a few remaining concerns.*

Response 1: Thank you for taking the time to review our manuscript again. We sincerely appreciate your acknowledgment of our revisions and your positive feedback on the expanded “uncertainties” section.

Comment 2: *I still feel there are reproducibility issues with this dataset. The figshare link provided (<https://figshare.com/s/6741ba3d1cf7f64d64ad>) has ‘derived’ datasets used to generate figures and the R code used to create them, but where is the ‘raw’ enzyme data? I am expecting one row per observation compiled by the authors, with enzyme values pulled from published literature and ideally each row has a doi or citation for where that enzyme data came from. Apologies if I am missing it, but I don’ t see this key file. This seems critical for full traceability so that others could be able to verify or build upon the work.*

Response 2: Thank you very much for describing your expectations for data reproducibility. For the revised manuscript, we have now uploaded the primary dataset containing the compiled enzyme activity measurements, with each entry linked to its original source. This file includes one row per observation, the corresponding enzyme values, and the associated DOI for each data point extracted from the literature.

The dataset is available alongside the derived data and R code in the updated Data Availability section (<https://figshare.com/s/6741ba3d1cf7f64d64ad>). We believe this addition provides greater transparency and will facilitate further validation and potential extension of our study.

Comment 3: *I am also unclear where the stoichiometric ratios for microbial biomass and soil organic matter came from, critical for the ecoenzymatic model (equations 3-6). Did every paper with enzyme data provide these values? If not, where did they come from? Are these values provided in the file in figshare?*

Response 3: Most studies reporting enzyme activities also provided microbial biomass and soil organic matter C:N:P ratios, but a few papers did not measure all required variables (e.g., enzyme activity, microbial biomass C:N:P, and soil organic matter stoichiometry). We thus adopted a common approach used in large-scale synthesis studies: for datasets with missing stoichiometric values, we supplemented them with spatially explicit global databases. This method ensures robustness while avoiding data exclusion due to incomplete reporting.

We have described this data compilation strategy in detail in the Methods section. The specific global gridded datasets and sources used for each variable (e.g., SOC, TN, TP, MBC, MBN, MBP) are also comprehensively listed in the supplementary Table S1, which provides full citations and access information for all external data products.

“When original studies did not fully report all soil nutrient and microbial biomass parameters, missing indicators were extracted from global data maps using the geographical coordinates of the sampling sites. SOC and TN contents were extracted from SoilGrids database (<https://soilgrids.org>)⁵¹. A global map of TP content was taken from the Land-Atmosphere Interaction Research Group at Sun Yat-sen University (<http://global-change.bnu.edu.cn/research/soilw>)⁵². Global maps of MBC, MBN, MBP contents were from a previous study⁵³.” (Lines 439-446)

References:

51. Hengl, T. et al. SoilGrids250m: Global gridded soil information based on machine learning. *PLoS One* **12**, e0169748 (2017).
52. Shangquan, W., Hengl, T., Mendes de Jesus, J., Yuan, H. & Dai, Y. Mapping the global depth to bedrock for land surface modeling. *J. Adv. Model. Earth Syst.* **9**, 65–88 (2017).
53. Gao, D. et al. Three-dimensional mapping of carbon, nitrogen, and phosphorus in soil microbial biomass and their stoichiometry at the global scale. *Glob. Change Biol.* **28**, 6728–6740 (2022). (Lines 699-712)

Comment 4: *Regarding the definition of nutrient use efficiency (Abstract line 49), should it be ‘invested in’ instead of ‘released via’ mineralization? As Reviewer 3 points out, we know nitrogen mineralization rates (flux) are highest in the tropics, so if this number is a mineralized N quantity it must be larger at low latitude, which should lead to smaller (not larger) NUE values since it is the denominator. But this is the opposite of what you found. However, if we’re talking about *investment in* nitrogen mineralization enzymes, this definition would make more sense. To address the valid points raised by Reviewer 3, I think it helps to repeatedly mention throughout this paper that the enzymatic approach is about metabolic investment/strategy, not fluxes or process rates per se. This would be good to highlight more clearly, that nutrient use efficiency is not the same as nutrient limitation (and what we learn from studying the former that’s different from the latter).*

Response 4: Thank you for the suggestion. We agree that “invested in” is a more precise description, and we have modified the text accordingly in the Abstract section. In addition, as you suggested, we have revised the manuscript to consistently emphasize that the coenzymatic stoichiometry approach used here reflects microbial metabolic investment and resource allocation strategy, not the indirect measurements of in situ nutrient cycling

fluxes.

“While nutrient use efficiency of soil microorganisms, the proportion of assimilated nutrients allocated into biosynthesis rather than invested in mineralization, is a critical microbial functional trait,” (Lines 48-51)

“Microbial NUE and PUE represent the proportion of these assimilated N and P allocated into biosynthesis (primarily growth) versus investment in mineralization pathways, reflecting the metabolic investment strategy between nutrient assimilation and potential mineralization⁵⁻⁷. Higher microbial NUE and PUE thus reflect more efficient allocation toward biomass production relative to mineralization investments^{5,7}.” (Lines 78-86)

“The estimated microbial NUE and PUE reflect the metabolic investment strategy, as reflected in microbial allocation toward C- vs. N- or P-acquiring coenzymes, linking microbial nutrient demand with soil nutrient availability. They thus capture microbial trade-offs in resource partitioning toward growth versus potential nutrient mineralization, rather than measuring actual *in-situ* process rates.” (Lines 154-160)

“Compared to the ¹⁸O method, the coenzymatic stoichiometric approach focuses on the enzymatic investment strategy employed by microorganisms in response to stoichiometric imbalances between microbial biomass and external substrates, emphasizing the enzymatic investment strategy to acquire limiting resources rather than directly quantifying mineralization fluxes¹⁴.” (Lines 368-374)

Comment 5: Line 52: seem to be missing some words in this sentence,

*should perhaps be ‘we predicted *the controls on* microbial nitrogen *use* efficiency...’ or similar. Not sensible as written.*

Response 5: Thank you for noting this point. We have modified this sentence in the revised manuscript.

“Here, we estimated microbial nitrogen use efficiency (NUE, $n = 2,012$) and phosphorus use efficiency (PUE, $n = 3,419$) across terrestrial ecosystems using the ecoenzymatic stoichiometric approach.” (Lines 53-56)

Comment 6: *Line 60: could you add a phrase for why NUE is lower at high latitude, same as you have for the PUE phrase?*

Response 6: We have added a phrase in the Abstract section that explains the likely mechanism underlying lower NUE in high-latitude regions. To comply with the journal’s strict word limit (150 words) for the Abstract, the explanation is brief.

“Spatial upscaling showed that tundra and boreal forest soils have markedly lower NUE than other regions, suggesting high nitrogen investments in nutrient acquisition in cold ecosystems,” (Lines 60-63)

Comment 7: *Lines 303-306: I do not understand how this paradox works. If higher C:N and C:P in the decomposing substrate forces more investment in nutrient acquiring enzymes, shouldn’t NUE and PUE values be lower? Can you clarify why not?*

Response 7: Thank you for identifying the unclear statement. In the revised manuscript, we have clarified that while increased coenzyme investment initially represents a metabolic cost, this investment is prioritized to acquire and retain scarce nutrients. While costly (lowering CUE), this strategy

minimizes nutrient loss, directly enabling higher cellular nutrient retention efficiency (NUE and PUE).

“When substrate C:nutrient ratios exceed microbial metabolic thresholds, microorganisms initially prioritize coenzyme production over biomass retention, increasing nutrient acquisition at the cost of growth, thereby decreasing NUE and PUE⁴⁷. While high substrate C:nutrient ratios force investment in coenzymes (a metabolic cost), this strategy allows microbes to minimize nutrient losses, thereby maintaining high cellular N and P retention (high NUE and PUE) despite the energetic cost to C-acquisition (low CUE).”

(Lines 324-331)

Response to Reviewer #3:

Comment 1: *I think that the authors have addressed my concerns adequately. They have substantially rewritten the manuscript. I would recommend accepting the paper.*

Response 1: Thank you very much for recommending the acceptance of our manuscript. We sincerely appreciate the time and effort you have dedicated to reviewing our work. We are also grateful for your constructive comments during the previous review round, which were invaluable in improving the paper's quality and clarity.